# IRAD: Implicit Representation-driven Image Resampling against Adversarial Attacks

**Yue Cao**[1,2]   **Tianlin Li**[2]   **Xiaofeng Cao**[3]   **Ivor Tsang**[1,2]   **Yang Liu**[2]   **Qing Guo**[1] *

[1] CFAR and IHPC, Agency for Science, Technology and Research (A*STAR), Singapore

[2] School of Computer Science and Engineering, Nanyang Technological University, Singapore

[3] Jilin University, China

## Abstract

We introduce a novel approach to counter adversarial attacks, namely, image resampling. Image resampling transforms a discrete image into a new one, simulating the process of scene recapturing or rerendering as specified by a geometrical transformation. The underlying rationale behind our idea is that image resampling can alleviate the influence of adversarial perturbations while preserving essential semantic information, thereby conferring an inherent advantage in defending against adversarial attacks. To validate this concept, we present a comprehensive study on leveraging image resampling to defend against adversarial attacks. We have developed basic resampling methods that employ interpolation strategies and coordinate shifting magnitudes. Our analysis reveals that these basic methods can partially mitigate adversarial attacks. However, they come with apparent limitations: the accuracy of clean images noticeably decreases, while the improvement in accuracy on adversarial examples is not substantial. We propose implicit representation-driven image resampling (IRAD) to overcome these limitations. First, we construct an implicit continuous representation that enables us to represent any input image within a continuous coordinate space. Second, we introduce SampleNet, which automatically generates pixel-wise shifts for resampling in response to different inputs. Furthermore, we can extend our approach to the state-of-the-art diffusion-based method, accelerating it with fewer time steps while preserving its defense capability. Extensive experiments demonstrate that our method significantly enhances the adversarial robustness of diverse deep models against various attacks while maintaining high accuracy on clean images. We released our code in https://github.com/tsingqguo/irad.

## 1 Introduction

Adversarial attacks can mislead powerful deep neural networks by adding optimized adversarial perturbations to clean images (Croce & Hein, 2020b; Kurakin et al., 2018; Goodfellow et al., 2014; Guo et al., 2020; Huang et al., 2023a), posing severe threats to intelligent systems. Existing works enhance the adversarial robustness of deep models by retraining them with the adversarial examples generated on the fly (Tramèr et al., 2018; Shafahi et al., 2019; Andriushchenko & Flammarion, 2020) or removing the perturbations before processing them (Liao et al., 2018; Huang et al., 2021b; Ho & Vasconcelos, 2022; Nie et al., 2022). These methods assume that the captured image is fixed. Nevertheless, in the real world, the observer can see the scene of interest several times via different observation ways since the real world is a continuous space and allows observers to resample the signal reflected from the scene. This could benefit the robustness of the perception system. We provide an illustrative example in Fig. 1 (a): when an image taken from a particular perspective is subjected to an attack that misleads the deep model (*e.g.*, ResNet50), by altering the viewing way, the same object can be re-captured and correctly classified. Such a process is also known as image resampling (Dodgson, 1992) that transforms a discrete image into a new one, simulating the process of scene recapturing or rerendering as specified by a geometrical transformation. In this work, we aim to study using image resampling to enhance the adversarial robustness of deep models against adversarial attacks.

---

*Qing Guo is the corresponding author (tsingqguo@ieee.org)

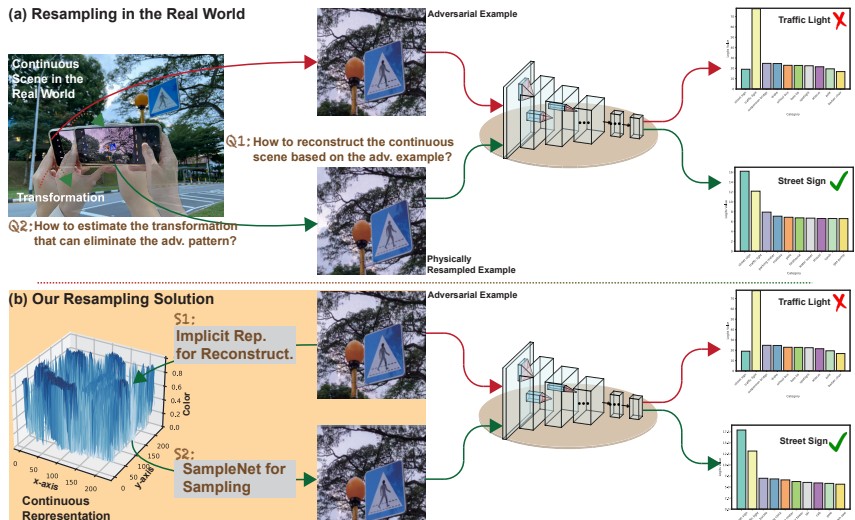

**Figure 1:** (a) shows the pipeline of resampling in the real world and the corresponding predictions, which inspires our main idea. Two main questions must be solved for the resampling-based solution (*i.e.*, Q1 and Q2). (b) shows the pipeline of the proposed method and two solutions (*i.e.*, S1 and S2) to address the two questions.

To achieve the image resampling with a given adversarial example, we must address two key questions: how to reconstruct the continuous scene based on the input? And how do we estimate the transformation that can eliminate the adversarial perturbation effectively? Note that, with a single-view input, we can only simulate the 2D transformations. To this end, we first provide a general formulation for the resampling-based adversarial defense. With this formulation, we design several basic resampling methods and conduct a comprehensive study to validate their effectiveness and limitations for adversarial defense. Then, we propose an implicit representation-driven resampling method (IRAD). Specifically, we first construct an implicit continuous representation for reconstruction, which enables us to represent any input image within a continuous coordinate space. Second, we introduce SampleNet, which automatically generates pixel-wise shifts for resampling in response to different inputs. Furthermore, we can extend our approach to the state-of-the-art diffusion-based method, accelerating it with fewer time steps while preserving its defense capability. We conduct extensive experiments on public datasets, demonstrating that our method can enhance the adversarial robustness significantly while maintaining high accuracy on clean images.

## 2 RELATED WORK

**Image Resampling.** Resampling is transforming a discrete image, defined at one set of coordinate locations, to a new set of coordinate points. Resampling can be divided conceptually into two processes: reconstructing the discrete image to a continuous image and then sampling the interpolated image (Parker et al., 1983). Among the existing reconstruction functions, nearest neighbor interpolation and bilinear interpolation are the most frequently adopted (Han, 2013/03). Nearest neighbor interpolation assigns the value of the nearest existing pixel to the new pixel coordinate, whereas bilinear interpolation calculates the new pixel value by taking a weighted average of the surrounding pixels in a bilinear manner. The resampling process involves refactoring pixels in the input image, which allows us to explore new approaches for mitigating adversarial attacks. Our paper investigates the potential of resampling to break malicious textures from adversarial inputs, which has not been studied in the community.

**Adversarial Attack and Defense.** White box attacks assume the attacker has full knowledge of the target model, including its architecture, weights, and hyper-parameters. This allows the attacker to generate adversarial examples with high fidelity using gradient-based optimization techniques, such as FGSM (Goodfellow et al., 2014), BIM (Kurakin et al., 2018), PGD (Madry et al., 2017), and others (Huang et al., 2023b). Other attacks also include black box attacks like Square Attack (Andriushchenko et al., 2020) and patch-wise attacks (Gao et al., 2020), as well as transferability-based attacks (Liu et al., 2016; Wang & He, 2021; Wang et al., 2021). AutoAttack (Croce & Hein, 2020b) has been proposed as a more comprehensive evaluation framework for adversarial attacks. AutoAttack combines several white box and black box attacks into a single framework and evaluates the robustness of a model against these attacks.

Adversarial defense can be categorized into two main types: adversarial training and adversarial purification (Nie et al., 2022). Adversarial training involves incorporating adversarial samples during the training process (Goodfellow et al., 2014; Madry et al., 2017; Athalye et al., 2018; Rade & Moosavi-Dezfooli, 2021; Ding et al., 2018; Zhang et al., 2020a; Jia et al., 2022b), and training with additional data generated by generative models (Sehwag et al., 2021). On the other hand, adversarial purification functions as a separate defense module during inference and does not require additional training time for the classifier (Guo et al., 2017; Xu et al., 2017; Sun et al., 2019; Ho & Vasconcelos, 2022).

## 3  IMAGE RESAMPLING (IR) AGAINST ADVERSARIAL ATTACK

### 3.1  PROBLEM FORMULATION

Given a dataset $\mathcal{D}$ with the data sample $\mathbf{X} \in \mathcal{X}$ and its label $y \in \mathcal{Y}$, the deep supervised learning model tries to learn a mapping or classification function $\mathrm{F}(\cdot) : \mathcal{X} \rightarrow \mathcal{Y}$. The model $\mathrm{F}(\cdot)$ could be different deep architectures. Existing works show that deep neural networks are vulnerable to adversarial perturbations (Madry et al., 2017). Specifically, a clean input $\mathbf{X}$, an adversarial attack is to estimate a perturbation which is added to the $\mathbf{X}$ and can mislead the $\mathrm{F}(\cdot)$,

$$\mathrm{F}(\mathbf{X}') \neq y, \text{ subject to } \|\mathbf{X} - \mathbf{X}'\| < \epsilon \tag{1}$$

where $\| \cdot \|$ is a distance metric. Commonly, $\| \cdot \|$ is measured by the $L_p$-norm ($p \in \{1, 2, \infty\}$), and $\epsilon$ denotes the perturbation magnitude. We usually name $\mathbf{X}'$ as the adversarial example of $\mathbf{X}$. There are two ways to enhance the adversarial robustness of $\mathrm{F}(\cdot)$. The first is to retrain the $\mathrm{F}(\cdot)$ with the adversarial examples estimated on the fly during training. The second is to process the input and remove the perturbation during testing. In this work, we explore a novel testing-time adversarial defense strategy, *i.e.*, IMAGE RESAMPLING, to enhance the adversarial robustness of deep models.

**Image resampling (IR).** Given an input discrete image captured by a camera in a scene, image resampling is to simulate the re-capture of the scene and generate another discrete image (Dodgson, 1992). For example, we can use a camera to take two images in the same environment but at different time stamps (See Fig. 1). Although the semantic information within the two captured images is the same, the details could be changed because the hands may shake, the light varies, the camera configuration changes, *etc*. Image resampling uses digital operations to simulate this process and is widely used in the distortion compensation of optical systems, registration of images from different sources with one another, registration of images for time-evolution analysis, television and movie special effects, *etc*. In this work, we propose to leverage image resampling for adversarial defense, and the intuition behind this idea is that resampling in the real world could keep the semantic information of the input image while being unaffected by the adversarial textures (See Fig. 1). We introduce the naive implementation of IR in Sec. 3.2 and discuss the challenges for adversarial defense in Sec. 3.3.

### 3.2  NAIVE IMPLEMENTATION

Image resampling contains two components, *i.e.*, reconstruction and sampling. Reconstruction is to build a continuous representation from the input discrete image, and sampling generates a new discrete image by taking samples off the built representation (Dodgson, 1992). Specifically, given a discrete image $\mathbf{I} \in \mathbb{R}^{H \times W \times 3} = \mathbf{X}$ or $\mathbf{X}'$, we will design a reconstruction method to get the continuous representation of the input image, which can be represented as

$$\phi = \text{RECONS}(\mathbf{I}), \tag{2}$$

where $\phi$ denotes the continuous representation that can estimate the intensity or color of arbitrarily given coordinates that could be non-integer values; that is, we have

$$\mathbf{c}_{u,v} = \phi(u, v) \tag{3}$$

where $\mathbf{c}_{u,v}$ denotes the color of the pixel at the continuous coordinates $[u, v]$. With the reconstructed $\phi$, we sample the coordinates of all desired pixels and generate another discrete image by

$$\hat{\mathbf{I}}[i,j] = \mathbf{c}_{u_{i,j},v_{i,j}} = \phi(\mathbf{U}[i,j]), \text{ subject to, } \mathbf{U} = \mathbf{G} + \text{SAMPLER}. \tag{4}$$

where $\mathbf{G}$ and $\mathbf{U}$, both in $\mathbb{R}^{H \times W \times 2}$, share the same dimensions as the input and consist of two channels. The matrix $\mathbf{G}$ stores the discrete coordinates of individual pixels, i.e., $\mathbf{G}[i,j] = [i,j]$. Each element in $\mathbf{U}[i,j] = [u_{i,j}, v_{i,j}]$ denotes the coordinates of the desired pixel in a continuous representation. This will be placed at the $[i,j]$ location in the output and is calculated by adding the Sampler-predicted shift to $\mathbf{G}$.

By leveraging image resampling to simulate the re-capture of the interested scene for adversarial defense, we pose two requirements: ❶ The reconstructed continuous representation is designed to represent the interested scene in a continuous space according to the input adversarial example and should eliminate the effects of adversarial perturbation. ❷ The sampling process should break the adversarial texture effectively while preserving the main semantic information. Traditional or naive resampling methods can hardly achieve the above two goals.

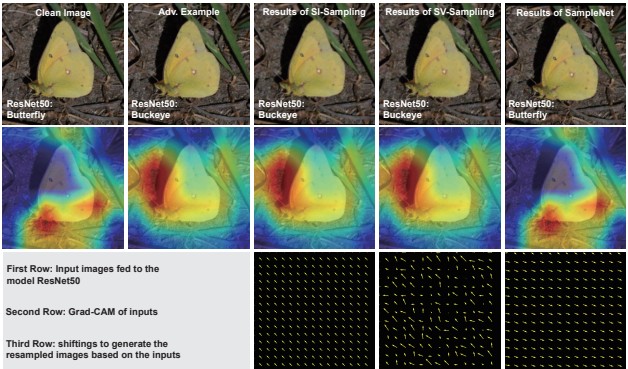

**Figure 2:** Comparison of different sampling strategies based on the bilinear interpolation as the reconstruction method.

**Image resampling via bilinear interpolation.** We can set the function RECONS in Eq. (2) in such a way: assigning the bilinear interpolation as the $\phi$ for arbitrary input images. Then, for an arbitrary given coordinates $[u,v]$, we formulate Eq. (3) as

$$\mathbf{c}_{u,v} = \phi(u,v) = \omega_1 \mathbf{I}[i_u^{-1}, j_v^{-1}] + \omega_2 \mathbf{I}[i_u^{-1}, j_v^{+1}] + \omega_3 \mathbf{I}[i_u^{+1}, j_v^{-1}] + \omega_4 \mathbf{I}[i_u^{+1}, j_v^{+1}] \tag{5}$$

where $[i_u^{-1}, j_v^{-1}]$, $[i_u^{-1}, j_v^{+1}]$, $[i_u^{+1}, j_v^{-1}]$, and $[i_u^{+1}, j_v^{+1}]$ are the four neighboring pixels around $[u,v]$ in the image $\mathbf{I}$ and $\{\omega_1, \omega_2, \omega_3, \omega_4\}$ are the bilinear weights that are calculated through four coordinates and are used to aggregate the four pixels.

**Image resampling via nearest interpolation.** Similar to bilinear interpolation, we can set the function RECONS in Eq. (2) in such a way assigning the nearest interpolation as the $\phi$ for arbitrary input images. Then, for an arbitrary given coordinates $[u,v]$, we formulate Eq. (3) as

$$\mathbf{c}_{u,v} = \phi(u,v) = \mathbf{I}[i_u, j_v], \tag{6}$$

where $[i_u, j_v]$ is the integer coordinate nearest the desired coordinates $[u,v]$.

For both interpolation methods, we can set naive sampling strategies as SAMPLER function: ❶ Spatial-invariant (SI) sampling, that is, we shift all raw coordinates along a fixed distance $d$ (See 3rd column in Fig. 2):

$$\mathbf{U} = \mathbf{G} + \text{SAMPLER}(d), \text{ subject to, } \mathbf{U}[i,j] = \mathbf{G}[i,j] + [d,d]. \tag{7}$$

❷ Spatial-variant (SV) sampling, that is, we randomly sample a shifting distance $r$ for the raw coordinates

$$\mathbf{U} = \mathbf{G} + \text{SAMPLER}(\gamma), \text{ subject to, } \mathbf{U}[i,j] = \mathbf{G}[i,j] + [d_1, d_2], d_1, d_2 \in \mathcal{U}(0,\gamma), \tag{8}$$

where $\mathcal{U}(0,\gamma)$ is a uniform distribution with the minimum and maximum being 0 and $\gamma$, respectively (See 4th column in Fig. 2). We can set different ranges $\gamma$ to see the changes in adversarial robustness.

We can use the two reconstruction methods with different sampling strategies against adversarial attacks. We take the CIFAR10 dataset and the WideResNet28-10 (Zagoruyko & Komodakis, 2016) as examples. We train the WideResNet28-10 on CIFAR10 dataset and calculate the clean testing dataset's accuracy, also known as the standard accuracy (SA). Then, we conduct the AutoAttack (Croce & Hein, 2020b) against the WideResNet28-

**Table 1:** Comparison of naive image resampling strategies on CIFAR10 via AutoAttack ($\epsilon_\infty = 8/255$).

| Naive IR methods | Stand. Acc. | Robust Acc. | Avg. Acc. |
|---|---|---|---|
| w.o. IR | **94.77** | 0 | 47.39 |
| IR(bil, SAMPLER($d = 1.5$)) | 85.24 | 42.30 | 63.77 |
| IR(bil, SAMPLER($\gamma = 1.5$)) | 53.81 | 30.24 | 42.03 |
| IR(nea, SAMPLER($d = 1.5$)) | 94.68 | 0.93 | 47.81 |
| IR(nea, SAMPLER($\gamma = 1.5$)) | 56.47 | 26.27 | 41.37 |

10 on all testing examples and calculate the accuracy, denoted as the robust accuracy (RA). We employ image resampling to process the input, which can be the clean image or adversarial example, and the processed input is fed to the WideResNet28-10. Then, we can calculate the SA

and RA of WideResNet28-10 with image resampling. We evaluate the effectiveness of the two naive reconstruction methods, *i.e.*, bilinear interpolation and nearest interpolation with the sampling strategies defined in Eq. (7) and Eq. (8), which are denoted as IR(bil, SAMPLER($d$ or $\gamma$)) and IR(nea, SAMPLER($d$ or $\gamma$), respectively. More details about the dataset, the model architecture, and the adversarial attack AutoAttack are deferred to Sec. 6.

### 3.3 DISCUSSIONS AND MOTIVATIONS

Based on the findings presented in Table 1, the following observations emerge: ❶ Bilinear interpolation with SI-sampling (*i.e.*, IR(bil, SAMPLER($d = 1.5$))) significantly enhances robust accuracy, albeit at the expense of a modest reduction in standard accuracy. In contrast, using nearest interpolation with SI-sampling results in only marginal variations in SA and RA. ❷ For both interpolation methods, SV-sampling leads to notable increases in RA, while si-

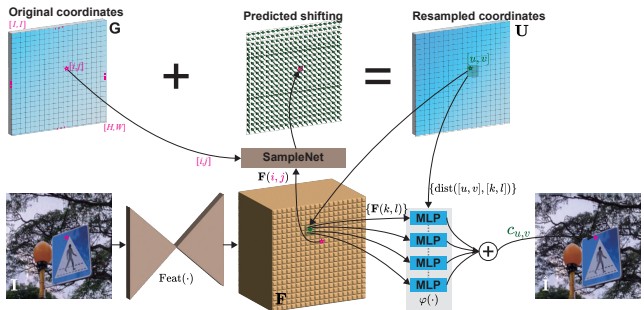

**Figure 3:** Pipeline of the proposed IRAD.

multaneously causing a significant reduction in SA. *Overall*, we see some effectiveness of leveraging naive image resampling methods against adversarial attacks. Nevertheless, such methods are far from being able to achieve high standard and robust accuracy at the same time. The reasons are that the naive interpolation-based reconstruction methods could not remove the perturbations while the sampling strategies are not designed to preserve the semantic information. As the example shown in Fig. 2, we feed the adversarial example to the bilinear interploration-based method with SI-sampling and SV-sampling, respectively, and use the Grad-CAM to present the semantic variations before and after resampling. Clearly, the two sampling strategy do not preserve the original semantic information properly. To address the issues, we propose a novel IR method in Sec. 4.

## 4 IMPLICIT CONTINUOUS REPRESENTATION-DRIVEN RESAMPLING

### 4.1 OVERVIEW

As analyzed in the previous section, we identified the limitations of naive resampling methods that can hardly achieve the two requirements for the reconstruction function and sampling function mentioned in Sec. 3.2. We propose implicit representation-driven image resampling (IRAD) to fill the gap. Specifically, we employ the implicit representation for the reconstruction function and train a SampleNet to automatically predict the shifting magnitudes according to different input images based on the built implicit representation. We display the whole process in Fig. 1 (b) and details in Fig. 3.

### 4.2 IMPLICIT REPRESENTATION

Given an input image $\mathbf{I} \in \mathbb{R}^{H \times W \times 3} = \mathbf{X}$ or $\mathbf{X}'$ that could be clean image or adversarial example, we first employ the local image implicit representation (Chen et al., 2021) to construct the continuous representation for the input image. Specifically, we calculate the pixel-wise embedding of the $\mathbf{I}$ via a deep model and get $\mathbf{F} = \text{FEAT}(\mathbf{I})$ where $\mathbf{F}(k, l)$ denotes the embedding of the pixel $[k, l]$. Given a desired coordinate $[u, v]$, we predict the color of $[u, v]$ based on the $\mathbf{F}(k, l)$ and the spatial distance between $[k, l]$ and $[u, v]$, that is, we can formulate the Eq. (3) as

$$c_{u,v} = \phi(u, v) = \sum_{[k,l] \in \mathcal{N}_{u,v}} \omega_{k,l} \varphi(\mathbf{F}(k, l), \text{dist}([u, v], [k, l])), \tag{9}$$

where $\mathcal{N}_{u,v}$ is a pixel set that contains the neighboring pixels around $[u, v]$, and the function $\text{dist}(\cdot)$ is to measure the spatial distance between $[u, v]$ and $[k, l]$. The function $\varphi(\cdot)$ is a multilayer perceptron (MLP) and predicts the color of the pixel $[u, v]$ according to the embedding of the pixel $[k, l]$ and their spatial distance. The key problem becomes how to train the deep model $\text{FEAT}(\cdot)$ and the MLP. In this work, we study four prediction tasks to train models:

- **Clean2Clean**. Given a clean image $\mathbf{I}$, we can reconstruct each pixel by feeding its raw coordinates to Eq. (9) and get the reconstruction $\hat{\mathbf{I}}$. Then, we use $L_1(\mathbf{I}, \hat{\mathbf{I}})$ to train the FEAT$(\cdot)$ and MLP.
- **Super-resolution**. We can also train the model via the super-resolution task as done in (Chen et al., 2021). That is, given a low-resolution input $\mathbf{I}_{\text{lr}}$ that is downsampled from a clean image $\mathbf{I}$, we aim to generate a higher resolution by sampling more coordinates and feeding them to Eq. (9), thus we can get a large-size image $\hat{\mathbf{I}}_{\text{hr}}$. We can also use the loss function $L_1(\mathbf{I}, \hat{\mathbf{I}}_{\text{hr}})$ to train the model.
- **Inpainting**. We generate a corrupted image $\mathbf{I}_{\text{mask}}$ by masking the clean image $\mathbf{I}$, we use Eq. (9) to restore the missing contents and get $\hat{\mathbf{I}}_{\text{mask}}$. We can also use the $L_1(\mathbf{I}, \hat{\mathbf{I}}_{\text{mask}})$ loss to train the model.
- **Gaussian denoising & Adversarial denoising**. We add clean images with random Gaussian noise or adversarially generated noise. Thus, we can get $\mathbf{I}_{\text{noise}}$. We aim to remove the noise via Eq. (9), and train the model via $L_1(\mathbf{I}, \hat{\mathbf{I}}_{\text{noise}})$.

We can test the trained models on different training tasks for adversarial defense and find that the model trained with adversarial denoising performs the best. For more details, please see Sec. 6.3.

## 4.3 SAMPLENET

Instead of the heuristic sampling strategies in Sec. 3.2, we propose to automatically predict the pixel-wise shifting according to the embedding of the input image. Intuitively, we aim to train a network, *i.e.*, SampleNet, which can output the shifting for all pixels (*i.e.*, $\mathbf{U}$ in Eq. (4)) to eliminate the adversarial perturbation further effectively, that is, we formulate Eq. (4) as

$$\mathbf{U}(i,j) = \text{SAMPLER}(\mathbf{I}) = \mathbf{G}(i,j) + \text{SAMPLENET}(\mathbf{F}(i,j),[i,j]), \qquad (10)$$

where $\mathbf{F} = \text{FEAT}(\mathbf{I})$, $\mathbf{G}$ is a matrix containing the original coordinates, for example, $\mathbf{G}(1,1) = [1,1]$. SAMPLENET is an MLP that takes the feature of pixel $[i,j]$ and the coordinate values as input and predicts its shifting directly. After training the implicit representation, we fix the FEAT and $\varphi$ in Eq. (9) and train the SAMPLENET via the adversarial denoise loss function. We visually compare the naive sampling strategies and the SampleNet in Fig. 2. The deep model can predict correctly on the resampled adversarial example with our SampleNet, while other sampling strategies cannot.

## 4.4 IMPLEMENTATION DETAILS

We follow the recent work (Chen et al., 2021) and set the deep model in (Lim et al., 2017) as the FEAT$(\cdot)$ to extract the embedding of the $\mathbf{I}$. We set $\varphi(\cdot)$ and the SAMPLENET as a five-layer MLP, respectively. We utilize the adversarial denoising task (See discussion in Sec. 6.3 ) to train the models through a two-stage training strategy. Specifically, we first train the implicit representation (*i.e.*, FEAT$(\cdot)$ and $\varphi(\cdot)$) on the training dataset, and then we fix their weights and train the SAMPLENET for sampling. Note that we follow a black-box setup; that is, we train our model based on adversarial examples crafted from ResNet18 and test the effectiveness on other deep models (See experimental section). Please refer to the Appendix for other training details.

## 5 RELATIONSHIP AND EXTENSION TO SOTAS

In the following, we discuss the relationship between our method and DISCO (Ho & Vasconcelos, 2022), and we also present a naive extension of our method to DiffPure (Nie et al., 2022), which could speed up DiffPure five times with similar defense performance.

**Relationship to implicit representation-based method (*e.g.*, (Ho & Vasconcelos, 2022)).** (Ho & Vasconcelos, 2022) employ implicit representation (Chen et al., 2021) to remove the adversarial perturbation and can enhance the robust accuracy under attacks significantly while preserving the high accuracy on the clean images. Different to (Ho & Vasconcelos, 2022), we employ the implicit representation (Chen et al., 2021) as a part of image resampling and study the influences of different training tasks. More importantly, our method contains the SampleNet that can automatically predict the suitable pixel-wise shifting and further recover the semantic information. We demonstrate the effectiveness of our method over (Ho & Vasconcelos, 2022) in Sec. 6.1.

**Extension to diffusion-based method (*e.g.*, (Nie et al., 2022)).** DiffPure (Nie et al., 2022) utilizes the diffusion model to purify the adversarial perturbation, which presents impressive results even though DiffPure is involved in the attacking pipeline. Nevertheless, DiffPure adopts large time steps

**Table 2:** Comparison on CIFAR10, CIFAR100 and ImageNet via AutoAttack ($\epsilon_\infty = 8/255$ for CIFAR10 and CIFAR100, $\epsilon_\infty = 4/255$ for ImageNet). "–" indicates no corresponding pre-trained model in the original paper.

| | Cifar10 | | | Cifar100 | | | ImageNet | | |
|---|---|---|---|---|---|---|---|---|---|
| Defense | SA | RA | Avg. | SA | RA | Avg. | SA | RA | Avg. |
| w.o. Defense | 94.77 | 0 | 47.39 | 81.66 | 3.48 | 42.57 | 76.72 | 0 | 38.36 |
| Bit Reduction (Xu et al., 2017) | 92.66 | 1.05 | 46.86 | 74.47 | 6.56 | 40.52 | 73.74 | 1.88 | 37.81 |
| Jpeg (Dziugaite et al., 2016) | 83.66 | 50.79 | 67.23 | 60.87 | 38.36 | 49.62 | 73.28 | 33.96 | 53.62 |
| Randomization (Xie et al., 2017) | 93.87 | 6.86 | 50.37 | 78.7 | 10.25 | 44.48 | 74.04 | 19.81 | 46.93 |
| Median Filter | 79.66 | 42.54 | 61.10 | 57.32 | 31.18 | 44.25 | 71.66 | 17.59 | 44.63 |
| NRP (Naseer et al., 2020) | 92.89 | 3.82 | 48.36 | 77.17 | 12.67 | 44.92 | 72.52 | 20.40 | 46.46 |
| STL (Sun et al., 2019) | 90.65 | 57.48 | 74.07 | – | – | – | 72.62 | 32.88 | 52.75 |
| DISCO (Ho & Vasconcelos, 2022) | 89.25 | 85.63 | 87.44 | 72.58 | 68.52 | 70.55 | 72.66 | 68.26 | 70.46 |
| DiffPure (Nie et al., 2022) | 89.67 | 87.54 | 88.61 | – | – | – | 68.28 | 68.04 | 68.16 |
| IRAD | 91.70 | 89.72 | 90.71 | 76.01 | 72.49 | 74.25 | 72.14 | 71.60 | 71.87 |

**Table 3:** Comparison on CIFAR10, CIFAR100, and ImageNet via BPDA.

| | Cifar10 | | | Cifar100 | | | ImageNet | | |
|---|---|---|---|---|---|---|---|---|---|
| Defense | SA | RA | Avg. | SA | RA | Avg. | SA | RA | Avg. |
| w.o. Defense | 94.77 | 0.02 | 47.40 | 81.67 | 0.49 | 41.08 | 76.72 | 0.00 | 38.36 |
| Bit Reduction (Xu et al., 2017) | 92.66 | 0.40 | 46.53 | 74.47 | 0.62 | 37.55 | 73.74 | 0.00 | 36.87 |
| Jpeg (Dziugaite et al., 2016) | 83.65 | 5.50 | 44.58 | 60.87 | 5.79 | 33.33 | 73.28 | 0.04 | 36.66 |
| Randomization (Xie et al., 2017) | 94.05 | 34.81 | 64.43 | 78.85 | 21.03 | 49.94 | 74.06 | 27.92 | 50.99 |
| Median Filter | 79.66 | 25.12 | 52.39 | 57.32 | 11.52 | 34.42 | 71.66 | 0.02 | 35.84 |
| NRP (Naseer et al., 2020) | 92.89 | 0.27 | 52.39 | 77.17 | 0.46 | 38.82 | 72.52 | 0.02 | 36.27 |
| STL (Sun et al., 2019) | 90.65 | 2.10 | 46.38 | – | – | – | 72.62 | 0.02 | 36.32 |
| DISCO (Ho & Vasconcelos, 2022) | 89.25 | 22.60 | 55.93 | 72.58 | 16.90 | 44.74 | 72.44 | 0.34 | 36.39 |
| DiffPure (Nie et al., 2022) | 89.15 | 87.06 | 88.11 | – | – | – | 68.85 | 61.42 | 65.14 |
| IRAD | 91.70 | 74.32 | 83.01 | 76.00 | 62.78 | 69.39 | 72.14 | 71.12 | 71.63 |

(*i.e.*, 100 time steps) to achieve good results, which is time-consuming; each image requires 5 seconds to limit the influence of adversarial perturbations for ImageNet images. We propose to use our method to speed up DiffPure while preserving its effectiveness. Specifically, we use DiffPure with 20-time steps to process input images and feed the output to our method with implicit representation as the reconstruction method and SampleNet as the sampler. As demonstrated in Sec. 6.1, our method combined with DiffPure achieves 5 times faster than the raw DiffPure with similar standard accuracy and robust accuracy.

## 6 EXPERIMENTAL RESULTS

In this section, we conduct extension experiments to validate our method. It is important to note that the results presented in each case are averaged over three experiments to mitigate the influence of varying random seeds. These experiments were conducted using the AMD EPYC 7763 64-Core Processor with 1 NVIDIA A100 GPUs.

**Metrics.** We evaluate IRAD and baseline methods on both clean and their respective adversarial examples, measuring their performance in terms of standard accuracy (SA) and robustness accuracy (RA). Furthermore, we compute the average of SA and RA as a comprehensive metric.

**Table 4:** IRAD generalization across DNNs on CIFAR10.

| DNNs | SA | RA | Avg. |
|---|---|---|---|
| WRN28-10 | 91.70 | 89.72 | 90.71 |
| WRN70-16 | 93.17 | 91.66 | 92.42 |
| VGG16_bn | 91.53 | 90.50 | 91.02 |
| ResNet34 | 91.16 | 88.82 | 89.99 |

**Datasets.** During training and evaluation of IRAD, We use three datasets: CIFAR10 (Krizhevsky et al., a), CIFAR100 (Krizhevsky et al., b) and ImageNet (Deng et al., 2009). To make training datasets of the same size, we randomly select 50000 images in ImageNet, which includes 50 samples for each class. IRAD is trained on pairs of adversarial and clean training images generated by the PGD attack (Madry et al., 2017) on ResNet18 (He et al., 2016). The PGD attack uses an $\epsilon$ value of 8/255 and 100 steps, with a step size 2/255. The SampleNet is also trained using adversarial-clean pairs generated by the PGD attack. In this part, we mainly present part of the results; other results will be shown in the Appendix.

**Attack scenarios.** ❶ *Oblivious adversary scenario:* We follow setups in RobustBench (Croce et al., 2021) and use the AutoAttack as the main attack method for the defense evaluation since AutoAttack is an ensemble of several white-box and black-box attacks, including two kinds of PGD attack, the FAB attack (Croce & Hein, 2020a), and the square attack (Andriushchenko et al., 2020), allowing for a more comprehensive evaluation. In addition to AutoAttack, we also report the results against FGSM (Goodfellow et al., 2014), BIM (Kurakin et al., 2018), PGD (Madry et al., 2017), RFGSM (Tramèr et al., 2018), TPgd (Zhang et al., 2019b), APgd (Croce & Hein, 2020b), EotPgd (Liu et al., 2018), FFgsm (Wang et al., 2020), MiFgsm (Dong et al., 2018), and Jitter (Schwinn et al., 2021).

We evaluate methods by utilizing adversarial examples generated through victim models that are pre-trained on clean images. ❷ *Adaptive adversary scenario:* We consider a challenging scenario where attackers know both the defense methods and classification models. Compared to the oblivious adversary, this is particularly challenging, especially regarding test-time defense (Athalye et al., 2018; Sun et al., 2019; Tramer et al., 2020). To evaluate the performance, we utilize the BPDA method (Athalye et al., 2018) that can circumvent defenses and achieve a high attack success rate, particularly against test-time defenses. ❸ *AutoAttack-based Adaptive adversary scenario.* We regard the IRAD and the target model as a whole and use AutoAttack to attack the whole process.

**Baelines.** We compare with 7 representative testing-time adversarial defense methods as shown in Table 2 including 2 SOTA methods, *i.e*., DISCO (Ho & Vasconcelos, 2022) and DiffPure (Nie et al., 2022). Note that, we run all methods by ourselves with their released models for a fair comparison. We also report more comparisons with training-time methods in the Appendix.

**Victim models.** We use adversarial attacks against the WideResNet28-10 (WRN28-10) on CIFAR10 and CIFAR100 and against ResNet50 on ImageNet since we do not find pre-trained WRN28-10 available for ImageNet. Then, we employ compared methods for defense for the main comparison study. Note that our model is trained with the ResNet18, which avoids the overfitting risk on the victim model. In addition, we also report the effectiveness of our model against other deep models.

## 6.1 Comparing with SOTA Methods

**Oblivious adversary scenario.** We compare IRAD and baseline methods across the CIFAR10, CIFAR100, and ImageNet datasets. With Table 2, we observe that: ❶ IRAD achieves the highest RA among all methods, especially when compared to the SOTA methods (e.g., DISCO (Ho & Vasconcelos, 2022) and DiffPure (Nie et al., 2022)). This demonstrates the primary advantages of the proposed method for enhancing adversarial robustness. ❷ IRAD is also capable of preserving high accuracy on clean data with only a slight reduction in SA compared to 'w.o. defense'. ❸ IRAD achieves the highest average accuracy among all methods across the three datasets. This demonstrates the effectiveness of the proposed method in achieving a favorable trade-off between SA and RA.

❹ Following the relationship between DISCO and IRAD as discussed in Sec. 5, the substantial advantages of IRAD over DISCO underscore the effectiveness of the proposed SampleNet. **Adaptive adversary scenario.** We further study the performance of all methods under a more challenging scenario where the attack is aware of both the defense methods and classification models and employs BPDA (Athalye

**Table 5:** Comparison on CIFAR10 via AutoAttack-based adaptive adversary.

|  | SA | RA | Avg. | Cost (ms) |
|---|---|---|---|---|
| DiffPure | 89.73 | 75.12 | 82.43 | 132.8 |
| DISCO | 89.25 | 0 | 44.63 | 0.38 |
| IRAD | 91.70 | 0 | 45.85 | 0.68 |
| DiffPure (t=20) | 93.66 | 8.01 | 50.83 | 27.25 |
| IRAD+DiffPure (t=20) | 93.42 | 74.05 | 83.74 | 27.70 |

et al., 2018) as the attack. As shown in Table 3, we have the following observations: ❶ RAs of all methods reduce under the BPDA. For example, DISCO gets 85.63% RA against AutoAttack but only achieves 22.60% RA under BPDA on CIFAR10. The RA of our method reduces from 89.72% to 74.32% on CIFAR10. ❷ Our method achieves the highest RA among all compared methods on all three datasets, with the exception of DiffPure in Cifar10. Although the RA of our method reduces, the relative improvements over other methods become much larger. DiffPure's high performance comes at the expense of significant computational resources. Therefore, we suggest a hybrid approach that combines the strengths of DiffPure and IRAD, as detailed in the following section. ❸ Our method still achieves the highest average accuracy when compared to all other methods, except for DiffPure in CIFAR-10, highlighting the advantages of IRAD in achieving a favorable trade-off between SA and RA under the adaptive adversary scenario. **AutoAttack-based Adaptive adversary scenario.** We also explored a tough defense scenario where attackers have full knowledge of defense methods and models. In Table 5, we found that DISCO and our original IRAD didn't improve robustness with zero RA. DiffPure maintains high SA and RA under adaptive settings but takes 132 ms per image, which is time-consuming. To speed up DiffPure, we can use fewer time steps, but this reduces RA significantly. When we combined IRAD with DiffPure (See Sec. 5), our IRAD+DiffPure (t=20) method achieved comparable SA and RA with DiffPure, and it's about five times faster. More experiments comparing different steps in the integration of IRAD and DiffPure are in Appendix A.6.

## 6.2 Generalization across DNNs, Datasets, and Other Attacks

Our IRAD is trained with the adversarial examples crafted from ResNet18 on respective datasets and we aim to test its generalization across other architectures and datasets. **Generalization across**

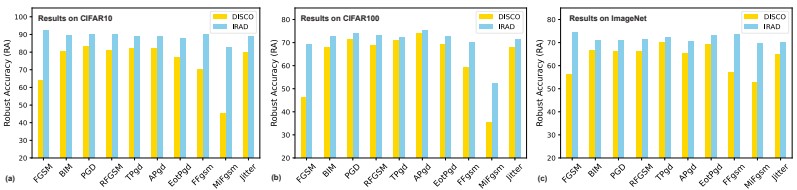

**Figure 4:** WRN28-10's robust accuracy under 10 attacks with DISCO and IRAD on three datasets.

**DNNs.** In Table 2, we test the ResNet18-trained IRAD on WRN28-10. Here, we further test it on the other three architectures, *i.e.*, WRN70-16, VGG16Bn, and ResNet34. As shown in Table 4, IRAD trained on ResNet18 could achieve similar SA and RA when we equip it to different DNNs, which demonstrates the high generalization of our method. **Generalization across Datasets.** We utilize the IRAD trained on one dataset to defend against attacks executed on a different dataset. For instance, we train IRAD on CIFAR10 with ResNet18 and employ it to counter AutoAttack when it attacks WRN28-10 on CIFAR100. As depicted in Table 6, our IRAD exhibits exceptional cross-dataset generalization. In other words, the IRAD model trained on CIFAR10 achieves comparable SA and RA as the IRAD model trained on CIFAR100, even when tested on the CIFAR100 dataset itself.

**Generalization against other attacks.** We further use IRAD trained via PGD-based adversarial examples against 10 attacking methods and compare it with the SOTA method DISCO. As shown in Fig. 4, we have the following observations: ❶ In the CIFAR10 and ImageNet datasets, IRAD achieves similar RA across all attack methods, which demonstrates the high generalization of IRAD across diverse attacks and even unknown attacks. Regarding the CIFAR100 dataset, IRAD presents slightly lower RA under MiFGSM than other attacks. ❷ IRAD consistently achieves significantly higher robust accuracies (RAs) across all attack scenarios, underscoring its advantages. Detailed results and additional experiments on more attacks can be found in Appendix A.3 and A.4.

**Table 6:** IRAD trained on the dataset to defend the attacks on another dataset.

| Training Testing | CIFAR10 | CIFAR100 |
|---|---|---|
| CIFAR10 | 89.72 | 67.14 |
| CIFAR100 | 88.68 | 72.49 |

## 6.3 ABLATION STUDY

**Training strategies comparison.** As detailed in Sec. 4.2, we can set different tasks to pre-train the IRAD. In this subsection, we compare the results of IRADs trained with different tasks on CIFAR10 datasets against AutoAttack. As shown in Table 7, we see that: ❶ IRAD trained with PGD-based denoising achieves the highest RA and average accuracy among all variants. IRAD with Gaussian denoising task gets the second best results but the RA reduces significantly. ❷ IRAD trained with other tasks has higher SAs than IRAD with denoising task but their RAs are close to zero. **Sampling strategies comparison.** We replace the SampleNet of IRAD with two naive sampling strategies introduced in Sec. 3.2 and evaluate the performance on the CIFAR10 dataset via AutoAttack ($\epsilon_{\infty} = 8/255$). As shown in Table 8, we see that SampleNet can achieve the highest SA, RA, and average accuracy, which demonstrates the effectiveness of our SampleNet. Additional results regarding the ablation study and the effectiveness analysis can be found in the Appendix.

**Table 7:** Comparison of implicit representation training strategies on Cifar10.

| Training Strategy | SA | RA | Avg. |
|---|---|---|---|
| Clean2Clean | **93.09** | 0.10 | 46.60 |
| Super Resolution | 93.05 | 0.20 | 46.63 |
| Restoration | 93.06 | 0.16 | 46.61 |
| Denoising (Gaussian) | 89.75 | 24.40 | 57.08 |
| Denoising (PGD) | 89.59 | **76.69** | **83.14** |

## 7 CONCLUSION

We have identified a novel adversarial defense solution, *i.e.*, image resampling, which can break the adversarial textures while maintaining the main semantic information in the input image. We provided a general formulation for the image resampling-based adversarial defense and designed several naive defensive resampling methods. We further studied the effectiveness and limitations of these naive methods. To fill the limitations, we proposed the implicit continuous representation-driven image resampling method by building the implicit representation and designing a SampleNet that can predict coordinate shifting magnitudes for all pixels according to different inputs. The experiments have demonstrated the advantages of our method over all existing methods. In the future, this method could be combined with the other two defensive methods, *i.e.*, denoising and adversarial training, for constructing much higher robust models.

**Table 8:** IRAD with different sampling strategies on CIFAR10.

| Sampling Strategies | SA | RA | Avg. |
|---|---|---|---|
| w.o. Sampling | 89.71 | 84.60 | 87.15 |
| SAMPLER($d = 1.5$) | 53.52 | 47.55 | 50.54 |
| SAMPLER($\gamma = 1.5$) | 82.19 | 77.90 | 80.05 |
| SampleNet | 91.70 | 89.72 | 90.71 |

ACKNOWLEDGMENT

This research is supported by the National Research Foundation, Singapore, and DSO National Laboratories under the AI Singapore Programme (AISG Award No: AISG2-GC-2023-008), and Career Development Fund (CDF) of the Agency for Science, Technology and Research (A*STAR) (No.: C233312028). Xiaofeng Cao is supported by the National Natural Science Foundation of China (Grant Number: 62206108). The research is also supported by the National Research Foundation, Singapore, and the Cyber Security Agency under its National Cybersecurity R&D Programme (NCRP25-P04-TAICeN). Any opinions, findings and conclusions or recommendations expressed in this material are those of the author(s) and do not reflect the views of the National Research Foundation, Singapore and Cyber Security Agency of Singapore.

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

# A APPENDIX

## A.1 EXPERIMENT SETTINGS IN DETAIL

In this section, we will provide a detailed description of the experiment settings.

For both implicit representation and SampleNet training of IRAD, we use pairs of clean and adversarial data. We generate image pairs under the PGD attack using ResNet18 for CIFAR10, CIFAR100 and ImageNet as the target model. The PGD attack employs an $\epsilon$ value of 8/255 and 100 steps, with a step size of 2/255.

For implicit representation training, we use Adam as the optimizer with a learning rate of 1e-4 and betas of (0, 0.9) as parameters. Throughout the training process, we use L1 loss to update the model, and the training is conducted with a batch size of 128.

For SampleNet training, we utilize cross-entropy loss, considering the prediction results of both clean and attacked images as the loss function. For optimization, we employ the Adam optimizer with a learning rate of 2e-4 for CIFAR10 and ImageNet, 1e-3 for CIFAR100, and betas set to (0, 0.9). The training of SampleNet is conducted with a batch size of 400 for CIFAR10, 200 for CIFAR100, and 8 for ImageNet.

## A.2 COMPARING WITH MORE SOTA METHODS ON CIFAR10 UNDER AUTOATTACK-BASED ADAPTIVE ADVERSARY SCENARIO

In this section, we present additional results from various methods comparison on Cifar10 under AutoAttack-based adaptive adversary scenario. As presented in Table 9, we present the baseline results from RobustBench, and compare them with the result we provided on Table 5 and Table 16.

From Table 9, we can conclude that IRAD+DiffPure method we proposed has advantages over existing SOTA methods on RobustBench.

**Table 9:** Comparison on CIFAR10 via AutoAttack ($\epsilon_\infty = 8/255$).

| Defense | SA | RA | Avg. | Classifier |
|---|---|---|---|---|
| w.o. Defense | 94.77 | 0 | 47.39 | WideResNet-28-10 |
| Robust_Peng (Peng et al., 2023) | 93.27 | 71.07 | 82.17 | RaWideResNet-70-16 |
| Better_Wang (Wang et al., 2023) | 93.25 | 70.69 | 81.97 | WideResNet-70-16 |
| Better_Wang (Wang et al., 2023) | 92.44 | 67.31 | 79.88 | WideResNet-28-10 |
| Fixing_Rebuffi (Rebuffi et al., 2021) | 92.23 | 66.58 | 79.41 | WideResNet-70-16 |
| Improving_Gowal (Gowal et al., 2021) | 88.74 | 66.11 | 77.43 | WideResNet-70-16 |
| Uncovering_Gowal (Gowal et al., 2020) | 91.1 | 65.88 | 78.49 | WideResNet-70-16 |
| Revisiting_Huang (Huang et al., 2022) | 91.58 | 65.79 | 78.69 | WideResNet-A4 |
| Fixing_Rebuffi (Rebuffi et al., 2021) | 88.5 | 64.64 | 76.57 | WideResNet-106-16 |
| Stable_Kang (Kang et al., 2021) | 93.73 | 71.28 | 82.51 | WideResNet-70-16, Neural ODE block |
| Fixing_Rebuffi (Rebuffi et al., 2021) | 88.54 | 64.25 | 76.40 | WideResNet-70-16 |
| Improving_Gowal (Gowal et al., 2021) | 87.5 | 63.44 | 75.47 | WideResNet-28-10 |
| Robustness_Pang (Pang et al., 2022) | 89.01 | 63.35 | 76.18 | WideResNet-70-16 |
| Helper_Rade(Rade & Moosavi-Dezfooli, 2021) | 91.47 | 62.83 | 77.15 | WideResNet-34-10 |
| Robust_Sehwag (Sehwag et al., 2021) | 87.3 | 62.79 | 75.05 | ResNest152 |
| Uncovering_Gowal (Gowal et al., 2020) | 89.48 | 62.8 | 76.14 | WideResNet-28-10 |
| Exploring_Huang (Huang et al., 2021a) | 91.23 | 62.54 | 76.89 | WideResNet-34-R |
| Exploring_Huang (Huang et al., 2021a) | 90.56 | 61.56 | 76.06 | WideResNet-34-R |
| Parameterizing_Dai (Dai et al., 2022) | 87.02 | 61.55 | 74.29 | WideResNet-28-10-PSSiLU |
| Robustness_Pang (Pang et al., 2022) | 88.61 | 61.04 | 74.83 | WideResNet-28-10 |
| Helper_Rade(Rade & Moosavi-Dezfooli, 2021) | 88.16 | 60.97 | 74.57 | WideResNet-28-10 |
| Fixing_Rebuffi (Rebuffi et al., 2021) | 87.33 | 60.75 | 74.04 | WideResNet-28-10 |
| Wider_Wu (Wu et al., 2021) | 87.67 | 60.65 | 74.16 | WideResNet-34-15 |
| Improving_Sridhar (Sridhar et al., 2022) | 86.53 | 60.41 | 73.47 | WideResNet-34-15 |
| Robust_Sehwag (Sehwag et al., 2021) | 86.68 | 60.27 | 73.48 | WideResNet-34-10 |
| Adversarial_Wu (Wu et al., 2020) | 88.25 | 60.04 | 74.15 | WideResNet-28-10 |
| Improving_Sridhar (Sridhar et al., 2022) | 89.46 | 59.66 | 74.56 | WideResNet-28-10 |

**Table 9 – continued from previous page**

| Defense | SA | RA | Avg. | Classifier |
|---|---|---|---|---|
| Geometry_Zhang (Zhang et al., 2020c) | 89.36 | 59.64 | 74.50 | WideResNet-28-10 |
| Unlabeled_Carmon (Carmon et al., 2019) | 89.69 | 59.53 | 74.61 | WideResNet-28-10 |
| Improving_Gowal (Gowal et al., 2021) | 87.35 | 58.63 | 72.99 | PreActResNet-18 |
| Towards_Addepalli (Addepalli et al., 2021) | 85.32 | 58.04 | 71.68 | WideResNet-34-10 |
| Efficient_Addepalli (Addepalli et al., 2022) | 88.71 | 57.81 | 73.26 | WideResNet-34-10 |
| Ltd_Chen (Chen & Lee, 2021) | 86.03 | 57.71 | 71.87 | WideResNet-34-20 |
| Helper_Rade(Rade & Moosavi-Dezfooli, 2021) | 89.02 | 57.67 | 73.35 | PreActResNet-18 |
| Adversarial_Jia (Jia et al., 2022a) | 85.66 | 57.61 | 71.64 | WideResNet-70-16 |
| Light_Debenedetti (Debenedetti et al., 2022) | 91.73 | 57.58 | 74.66 | XCiT-L12 |
| Light_Debenedetti (Debenedetti et al., 2022) | 91.3 | 57.27 | 74.29 | XCiT-M12 |
| Uncovering_Gowal (Gowal et al., 2020) | 85.29 | 57.2 | 71.25 | WideResNet-70-16 |
| Hydra_Sehwag(Sehwag et al., 2020) | 88.98 | 57.14 | 73.06 | WideResNet-28-10 |
| Helper_Rade(Rade & Moosavi-Dezfooli, 2021) | 86.86 | 57.09 | 71.98 | PreActResNet-18 |
| Ltd_Chen (Chen & Lee, 2021) | 85.21 | 56.94 | 71.08 | WideResNet-34-10 |
| Uncovering_Gowal (Gowal et al., 2020) | 85.64 | 56.86 | 71.25 | WideResNet-34-20 |
| Fixing_Rebuffi (Rebuffi et al., 2021) | 83.53 | 56.66 | 70.10 | PreActResNet-18 |
| Improving_Wang (Wang et al., 2020) | 87.5 | 56.29 | 71.90 | WideResNet-28-10 |
| Adversarial_Jia (Jia et al., 2022a) | 84.98 | 56.26 | 70.62 | WideResNet-34-10 |
| Adversarial_Wu (Wu et al., 2020) | 85.36 | 56.17 | 70.77 | WideResNet-34-10 |
| Light_Debenedetti (Debenedetti et al., 2022) | 90.06 | 56.14 | 73.10 | XCiT-S12 |
| Labels_Uesato (Uesato et al., 2019) | 86.46 | 56.03 | 71.25 | WideResNet-28-10 |
| Robust_Sehwag (Sehwag et al., 2021) | 84.59 | 55.54 | 70.07 | ResNet-18 |
| Using_Hendrycks (Hendrycks et al., 2019) | 87.11 | 54.92 | 71.02 | WideResNet-28-10 |
| Bag_Pang (Pang et al., 2020a) | 86.43 | 54.39 | 70.41 | WideResNet-34-20 |
| Boosting_Pang (Pang et al., 2020b) | 85.14 | 53.74 | 69.44 | WideResNet-34-20 |
| Learnable_Cui (Cui et al., 2021) | 88.7 | 53.57 | 71.14 | WideResNet-34-20 |
| Attacks_Zhang (Zhang et al., 2020b) | 84.52 | 53.51 | 69.02 | WideResNet-34-10 |
| Overfitting_Rice(Rice et al., 2020) | 85.34 | 53.42 | 69.38 | WideResNet-34-20 |
| Self_Huang (Huang et al., 2020) | 83.48 | 53.34 | 68.41 | WideResNet-34-10 |
| Theoretically_Zhang(Zhang et al., 2019b) | 84.92 | 53.08 | 69.00 | WideResNet-34-10 |
| Learnable_Cui (Cui et al., 2021) | 88.22 | 52.86 | 70.54 | WideResNet-34-10 |
| Adversarial_Qin (Qin et al., 2019) | 86.28 | 52.84 | 69.56 | WideResNet-40-8 |
| Efficient_Addepalli (Addepalli et al., 2022) | 85.71 | 52.48 | 69.10 | ResNet-18 |

**Table 9 – continued from previous page**

| Defense | SA | RA | Avg. | Classifier |
|---|---|---|---|---|
| Adversarial_Chen (Chen et al., 2020) | 86.04 | 51.56 | 68.80 | ResNet-50 |
| Efficient_Chen (Chen et al., 2022) | 85.32 | 51.12 | 68.22 | WideResNet-34-10 |
| Towards_Addepalli (Addepalli et al., 2021) | 80.24 | 51.06 | 65.65 | ResNet-18 |
| Improving_Sitawarin (Sitawarin et al., 2020) | 86.84 | 50.72 | 68.78 | WideResNet-34-10 |
| Robustness (Engstrom et al., 2019) | 87.03 | 49.25 | 68.14 | ResNet-50 |
| Harnessing_Kumari (Kumari et al., 2019) | 87.8 | 49.12 | 68.46 | WideResNet-34-10 |
| Metric_Mao (Mao et al., 2019) | 86.21 | 47.41 | 66.81 | WideResNet-34-10 |
| You_Zhang (Zhang et al., 2019a) | 87.2 | 44.83 | 66.02 | WideResNet-34-10 |
| Towards_Madry (Madry et al., 2017) | 87.14 | 44.04 | 65.59 | WideResNet-34-10 |
| Understanding_Andriushchenko(Andriushchenko & Flammarion, 2020) | 79.84 | 43.93 | 61.89 | PreActResNet-18 |
| Rethinking_Pang (Pang et al., 2019) | 80.89 | 43.48 | 62.19 | ResNet-32 |
| Fast_Wong (Wong et al., 2020) | 83.34 | 43.21 | 63.28 | PreActResNet-18 |
| Adversarial_Shafahi (Shafahi et al., 2019) | 86.11 | 41.47 | 63.79 | WideResNet-34-10 |
| Mma_Ding (Ding et al., 2018) | 84.36 | 41.44 | 62.90 | WideResNet-28-4 |
| Tunable_Kundu(Kundu et al., 2020) | 87.32 | 40.41 | 63.87 | ResNet-18 |
| Controlling_Atzmon (Atzmon et al., 2019) | 81.3 | 40.22 | 60.76 | ResNet-18 |
| Robustness_Moosavi (Moosavi-Dezfooli et al., 2019) | 83.11 | 38.5 | 60.81 | ResNet-18 |
| Defense_Zhang (Zhang & Wang, 2019) | 89.98 | 36.64 | 63.31 | WideResNet-28-10 |
| Adversarial_Zhang (Zhang & Xu) | 90.25 | 36.45 | 63.35 | WideResNet-28-10 |
| Adversarial_Jang(Jang et al., 2019) | 78.91 | 34.95 | 56.93 | ResNet-20 |
| Sensible_Kim (Kim & Wang, 2020) | 91.51 | 34.22 | 62.87 | WideResNet-34-10 |
| Adversarial_Zhang(Zhang & Xu) | 44.73 | 32.64 | 38.69 | 5-layer-CNN |
| Bilateral_Wang (Wang & Zhang, 2019) | 92.8 | 29.35 | 61.08 | WideResNet-28-10 |
| Enhancing_Xiao (Xiao et al., 2019) | 79.28 | 18.5 | 48.89 | DenseNet-121 |
| Manifold_Jin(Jin & Rinard, 2020) | 90.84 | 1.35 | 46.10 | ResNet-18 |
| Adversarial_Mustafa (Mustafa et al., 2019) | 89.16 | 0.28 | 44.72 | ResNet-110 |
| Jacobian_Chan (Chan et al., 2019) | 93.79 | 0.26 | 47.03 | WideResNet-34-10 |
| Clustr_Alfarra (Alfarra et al., 2020) | 91.03 | 0 | 45.52 | WideResNet-28-10 |
| IRAD+DiffPure (t=20) | 93.42 | 74.05 | 83.74 | WideResNet-28-10 |
| IRAD+DiffPure (t=25) | 91.42 | 77.71 | 84.57 | WideResNet-28-10 |
| IRAD+DiffPure (t=30) | 92.33 | 82.85 | 87.59 | WideResNet-28-10 |
| IRAD+DiffPure (t=35) | 89.14 | 84.57 | 86.86 | WideResNet-28-10 |
| IRAD+DiffPure (t=40) | 90.57 | 86.00 | 88.29 | WideResNet-28-10 |

## A.3 DETAILED DATA ON GENERALIZATION AGAINST OTHER ATTACKS

In this section, we showcase the outcomes of IRAD's performance when subjected to ten distinct attack scenarios on CIFAR10, CIFAR100, and ImageNet datasets, as illustrated in Table 10, Table 11, and Table 12. These results correspond to Fig. 4 in the main paper.

**Table 10:** WRN28-10 against 10 attacks on CIFAR10 ($\epsilon_\infty = 8/255$). **Table 11:** WRN28-10 against 10 attacks on CIFAR100 ($\epsilon_\infty = 8/255$). **Table 12:** ResNet50 against 10 attacks on ImageNet ($\epsilon_\infty = 4/255$).

| Attack | DISCO | IRAD |
|--------|-------|------|
| FGSM | 64.15 | **92.11** |
| BIM | 80.55 | **89.64** |
| PGD | 83.27 | **89.94** |
| RFGSM | 81.08 | **89.97** |
| TPgd | 82.13 | **88.77** |
| APgd | 82.10 | **89.09** |
| EotPgd | 76.83 | **87.56** |
| FFgsm | 70.41 | **90.23** |
| MiFgsm | 45.40 | **82.51** |
| Jitter | 79.64 | **89.01** |
| Avg. | 74.55 | **88.88** |

| Attack | DISCO | IRAD |
|--------|-------|------|
| FGSM | 46.38 | **69.36** |
| BIM | 67.99 | **72.45** |
| PGD | 71.25 | **74.12** |
| RFGSM | 68.67 | **73.04** |
| TPgd | 70.75 | **72.16** |
| APgd | 73.76 | **75.10** |
| EotPgd | 69.22 | **72.78** |
| FFgsm | 59.01 | **70.27** |
| MiFgsm | 35.38 | **52.09** |
| Jitter | 67.91 | **71.27** |
| Avg. | 63.03 | **70.26** |

| Attack | DISCO | IRAD |
|--------|-------|------|
| FGSM | 56.00 | **74.50** |
| BIM | 66.52 | **71.06** |
| PGD | 66.28 | **71.10** |
| RFGSM | 66.22 | **71.22** |
| TPgd | 70.00 | **72.12** |
| APgd | 65.44 | **70.66** |
| EotPgd | 69.30 | **73.28** |
| FFgsm | 57.06 | **73.50** |
| MiFgsm | 52.66 | **69.50** |
| Jitter | 64.66 | **70.24** |
| Avg. | 63.41 | **71.72** |

## A.4 PERFORMANCE UNDER TRANSFER-BASED ATTACKS

**Table 13:** WRN28-10 against VmiFgsm, VniFgsm, Admix attack on CIFAR10 ($\epsilon_\infty = 8/255$).

| Defense | VmiFgsm SA | VmiFgsm RA | VmiFgsm Avg. | VniFgsm SA | VniFgsm RA | VniFgsm Avg. | Admix SA | Admix RA | Admix Avg. |
|---------|----|----|------|----|----|------|----|----|------|
| No Defense | 94.77 | 0 | 47.39 | 94.77 | 0 | 47.39 | 94.77 | 3.82 | 49.30 |
| DISCO | 89.24 | 50.51 | 69.88 | 89.24 | 52.76 | 71.00 | 89.24 | 65.91 | 77.58 |
| DiffPure (t=100) | 89.21 | 85.55 | 87.38 | 89.21 | 85.49 | 87.35 | 89.21 | 85.23 | 87.22 |
| IRAD | 91.70 | 86.22 | 88.96 | 91.70 | 87.65 | 89.68 | 91.70 | 82.39 | 87.05 |

In this section, we conduct experiments on more transfer-based attacks (Wang & He, 2021; Wang et al., 2021), which include three attack methods (i.e., VmiFgsm, VniFgsm, and Admix). Our results demonstrate that our method significantly outperforms DISCO when facing all three attacks. As depicted in Table 13, in comparison to DiffPure, IRAD exhibits notably superior SA and RA when subjected to the VmiFgsm and VniFgsm attacks. While DiffPure slightly surpasses IRAD in RA under the Admix attack, it is worth mentioning that the inference process of DiffPure is approximately 200 times slower than IRAD as shown in Table 5.

**Table 14:** ResNet50 against VmiFgsm, VniFgsm, Admix attack on ImageNet ($\epsilon_\infty = 4/255$).

| Defense | VmiFgsm SA | VmiFgsm RA | VmiFgsm Avg. | VniFgsm SA | VniFgsm RA | VniFgsm Avg. | Admix SA | Admix RA | Admix Avg. |
|---------|----|----|------|----|----|------|----|----|------|
| No Defense | 76.72 | 0 | 38.36 | 76.72 | 0.04 | 38.38 | 76.72 | 3.32 | 40.02 |
| DISCO | 72.66 | 52.98 | 62.82 | 72.66 | 52.76 | 62.71 | 72.66 | 42.74 | 57.7 |
| DiffPure (t=150) | 69.42 | 64.9 | 67.16 | 69.42 | 64.22 | 66.82 | 69.42 | 61.34 | 65.38 |
| IRAD | 72.14 | 71.35 | 71.75 | 72.14 | 71.4 | 71.77 | 72.14 | 63.8 | 67.97 |

Furthermore, we conducted additional experiments on ImageNet, following the approach outlined in the DiffPure paper and setting t=150 when dealing with ImageNet.

Table 14 indicates that IRAD outperforms DISCO significantly across all three attack scenarios. Furthermore, not only does IRAD surpass DiffPure in performance under all three attacks, but it is also more than 100 times faster than DiffPure. This demonstrates IRAD's superior performance and time efficiency.

## A.5 ABLATION STUDY ON DIFFERENT RECONSTRUCTION METHODS

In this section, we conduct an ablation study to integrate various RECONS methods with SampleNet.

It's important to note that SampleNet is an MLP that takes the features of pixel [i,j] and the coordinate values as input, predicting their shifting directly. Since the Nearest and Bilinear RECONS methods could not generate the pixel features as the implicit representations do, we directly utilize the encoder from the implicit representations for a fair comparison. The experimental results are shown in Table 15.

**Table 15:** Comparing different reconstruction methods.

| RECONS | SA | RA | Avg. |
|---|---|---|---|
| Nearest | 93.46 | 0.96 | 47.21 |
| Bilinear | 92.61 | 79.20 | 85.91 |
| Implicit Representation | 91.70 | 89.72 | 90.71 |

As depicted in the table, despite having a high SA, Nearest RECONS lacks the capability to defend against adversarial attacks. This limitation arises from its ability to acquire only the pixel values originally sampled from the images. The Bilinear RECONS exhibits superior RA in comparison to the Nearest RECONS and outperforms basic image resampling methods under Bilinear RECONS showcased in Table 1 of the paper, highlighting the effectiveness of SampleNet. Compared to these two RECONS, the implicit representation RECONS achieves an even higher RA and Avg., demonstrating their superiority over these two RECONS. Overall, the data from the table suggests that Nearest and Bilinear RECONS are less effective than implicit representation-based methods when integrated with SampleNet.

## A.6 Comparing Different Steps in the Integration of IRAD and DiffPure

In this section, we conduct additional experiments on integrating IRAD with various steps of DiffPure. The results are presented in Table 16. Due to the significant time and resource costs associated with implementing DiffPure, we performed the experiment on 350 randomly selected images from the test set. The findings indicate that IRAD consistently enhances DiffPure's performance across various steps. For example, with t=40, the combination of IRAD and DiffPure (t=40) notably increases the Robust Accuracy (RA) of DiffPure (t=40) from 59.42% to 86.00%. Moreover, it surpasses DiffPure (t=100) in terms of SA, RA, Avg., and time efficiency, demonstrating significant improvements.

**Table 16:** Comparing different steps in the integration of IRAD and DiffPure.

| | SA | RA | Avg. | Cost (ms) |
|---|---|---|---|---|
| DiffPure | 89.73 | 75.12 | 82.43 | 132.80 |
| DISCO | 89.25 | 0 | 44.63 | 0.38 |
| IRAD | 91.70 | 0 | 45.85 | 0.68 |
| DiffPure (t=20) | 93.66 | 8.01 | 50.83 | 27.25 |
| IRAD+DiffPure (t=20) | 93.42 | 74.05 | 83.74 | 27.70 |
| DiffPure (t=25) | 93.55 | 18.28 | 55.92 | 31.93 |
| IRAD+DiffPure (t=25) | 91.42 | 77.71 | 84.57 | 32.41 |
| DiffPure (t=30) | 92.85 | 36.28 | 64.57 | 37.62 |
| IRAD+DiffPure (t=30) | 92.33 | 82.85 | 87.59 | 38.37 |
| DiffPure (t=35) | 93.50 | 48.75 | 71.13 | 44.18 |
| IRAD+DiffPure (t=35) | 89.14 | 84.57 | 86.86 | 44.86 |
| DiffPure (t=40) | 93.71 | 59.42 | 76.57 | 51.78 |
| IRAD+DiffPure (t=40) | 90.57 | 86.00 | 88.29 | 52.33 |

## A.7 The Effectiveness of Implicit Representation Reconstruction

In this part, we employ four metrics to assess the reconstruction of the implicit representation: PSNR (Hore & Ziou, 2010), SSIM (Wang et al., 2004), loss of low frequency and high frequency FFT (Fast Fourier Transform) (Cooley & Tukey, 1965), and LPIPS (Zhang et al., 2018).

**Table 17:** Image quality results on CIFAR 10.

| | Clean and Adv | Clean and RECONS(Adv) |
|---|---|---|
| PSNR | 32.10 | 36.01 |
| SSIM | 0.9936 | 0.9973 |
| Low Frequency FFT Loss | 0.0626 | 0.0506 |
| High Frequency FFT Loss | 0.3636 | 0.2489 |
| LPIPS | 0.0771 | 0.0169 |

1. PSNR: As shown in Table 17, the PSNR value is higher for implicit representation reconstructed images (36.01) compared to adversarial images (32.10). This indicates that the reconstructed adversarial images have higher fidelity and less distortion compared to the clean images after reconstruction.

2. SSIM: The SSIM value is higher for reconstructed adversarial images (0.9973) compared to adversarial images (0.9936). This suggests that the reconstructed adversarial images better preserve the structural details of the original images.

3. FFT: Both FFT Low Loss and FFT High Loss are lower for reconstructed adversarial images compared to clean images. This implies that less information is lost in frequency domains during the reconstruction of adversarial images.

4. LPIPS: The LPIPS value is significantly lower for reconstructed adversarial images (0.0169) compared to adversarial images (0.0771). This suggests that the reconstructed adversarial images are more perceptually similar to the original images compared to the adversarial images.

Overall, the table suggests that the reconstructed versions of adversarial images tend to exhibit better quality and closer resemblance to the original images across multiple metrics. This also implies that the implicit representation does not sacrifice meaningful information in the raw image when eliminating adversarial perturbations. This demonstrates the effectiveness of using implicit continuous representation to represent images within a continuous coordinate space.

**Table 18:** The comparison of feature distances.

| Comparison | Euclidean Distance |
|---|---|
| Clean and Adv | 2.91 |
| Clean and RECONS(Adv) | 1.21 |
| Clean and IRAD(Adv) | 0.75 |

### A.8 THE EFFECTIVENESS OF SAMPLENET

In this section, we introduce two additional perspectives to analyze our method and observe the following: First, SampleNet significantly diminishes semantic disparity, bringing reconstructed adversarial and clean representations into closer alignment. Second, SampleNet aligns the decision paths of reconstructed adversarial examples with those of clean examples.

Firstly, we delve into analyzing the features to comprehend how the semantics of the images evolve and perform PCA to visualize the analysis. We randomly selected 100 images from one class and obtained the features of clean images (Clean), adversarial images (Adv), adversarial images after implicit reconstruction (RECONS(Adv)), and adversarial images after IRAD (IRAD(Adv)). We computed the mean in each dimension of the features of clean images to

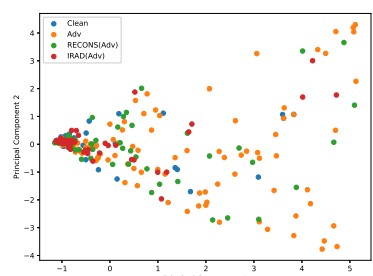

**Figure 5:** The PCA results.

establish the central point of these clean features. Subsequently, we computed the Euclidean distance between the features of other images and the center of clean images. As shown in Table 18, the average Euclidean distance of adversarial image features is significantly distant from clean image features. After the reconstruction of implicit representation, the adversarial image features become closer to the center of clean images' features. By utilizing IRAD, the distance becomes even closer. The PCA findings also illustrate a comparable phenomenon. As depicted in Fig. 5, clean adversarial images are notably distant from clean images. After the reconstruction of implicit representation, there are fewer samples situated far from clean images. Moreover, significantly fewer images lie outside the clean distribution after employing IRAD. Therefore, this demonstrates that the shift map generated by SampleNet can notably reduce semantic disparity and bring adversarial and clean representations into closer alignment.

Secondly, we follow the methodology of previous research to conduct decision rationale analysis through decision path analysis (Khakzar et al., 2021; Li et al., 2021; Wang et al., 2018; Qiu et al., 2019; Li et al., 2023; Xie et al., 2022), aiming to understand the decision-making process following resampling. We obtain the decision paths of Adv, RECONS(Adv), and IRAD(Adv), and calculate their cosine similarity with the

**Table 19:** Decision path comparison.

| Comparison | Similarity |
|---|---|
| Clean and Adv | 0.5482 |
| Clean and RECONS(Adv) | 0.9915 |
| Clean and IRAD(Adv) | 0.9987 |

decision path of clean images. As shown in Table 19, after the reconstruction of implicit representation, the decision path becomes more similar. With the usage of IRAD, the similarity becomes even closer. These results demonstrate that after resampling, our method exhibits a closer alignment with the decision rationale observed in clean images.

These two experiments further highlight the effectiveness of our resampling technique in disrupting adversarial textures and mitigating adversarial attacks.

### A.9 MORE VISUALIZATION RESULTS

In this section, we present some examples of applying IRAD on CIFAR10, CIFAR100, and ImageNet.

For each figure, (a) and (b) present a clean image and the corresponding adversarial counterpart as well as the predicted logits from the classifier. (c) and (d) show the results of using a randomly resampling strategy to handle the clean and adversarial images before feeding them to the classifier. (e) and (f) display the results of leveraging IRAD to handle the inputs.

The visualizations in Fig. 6, 7, and 8 reveal that the randomly resampled clean image produces almost identical logits to the raw clean image. On the other hand, the randomly resampled adversarial image yields lower confidence in the misclassified category, but it does not directly rectify prediction errors. IRAD, however, not only maintains the logits for clean images but also corrects classification errors and generates logits similar to clean images for adversarial images. This is the reason why IRAD enhances robustness while maintaining good performance on clean images.

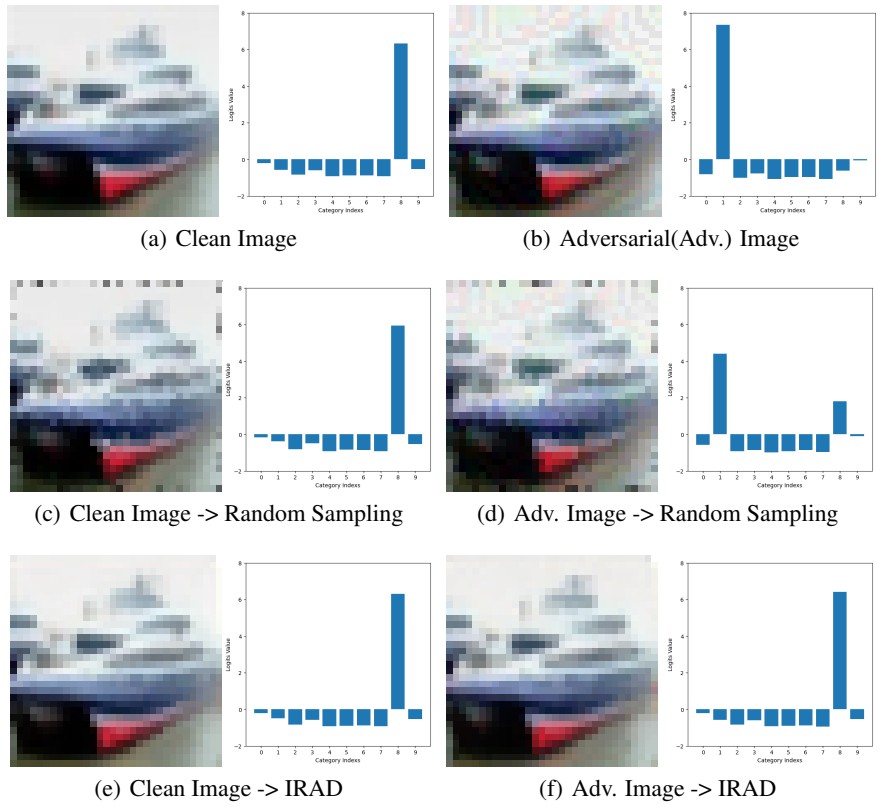

(a) Clean Image

(b) Adversarial(Adv.) Image

(c) Clean Image -> Random Sampling

(d) Adv. Image -> Random Sampling

(e) Clean Image -> IRAD

(f) Adv. Image -> IRAD

**Figure 6:** Case visualization of CIFAR10

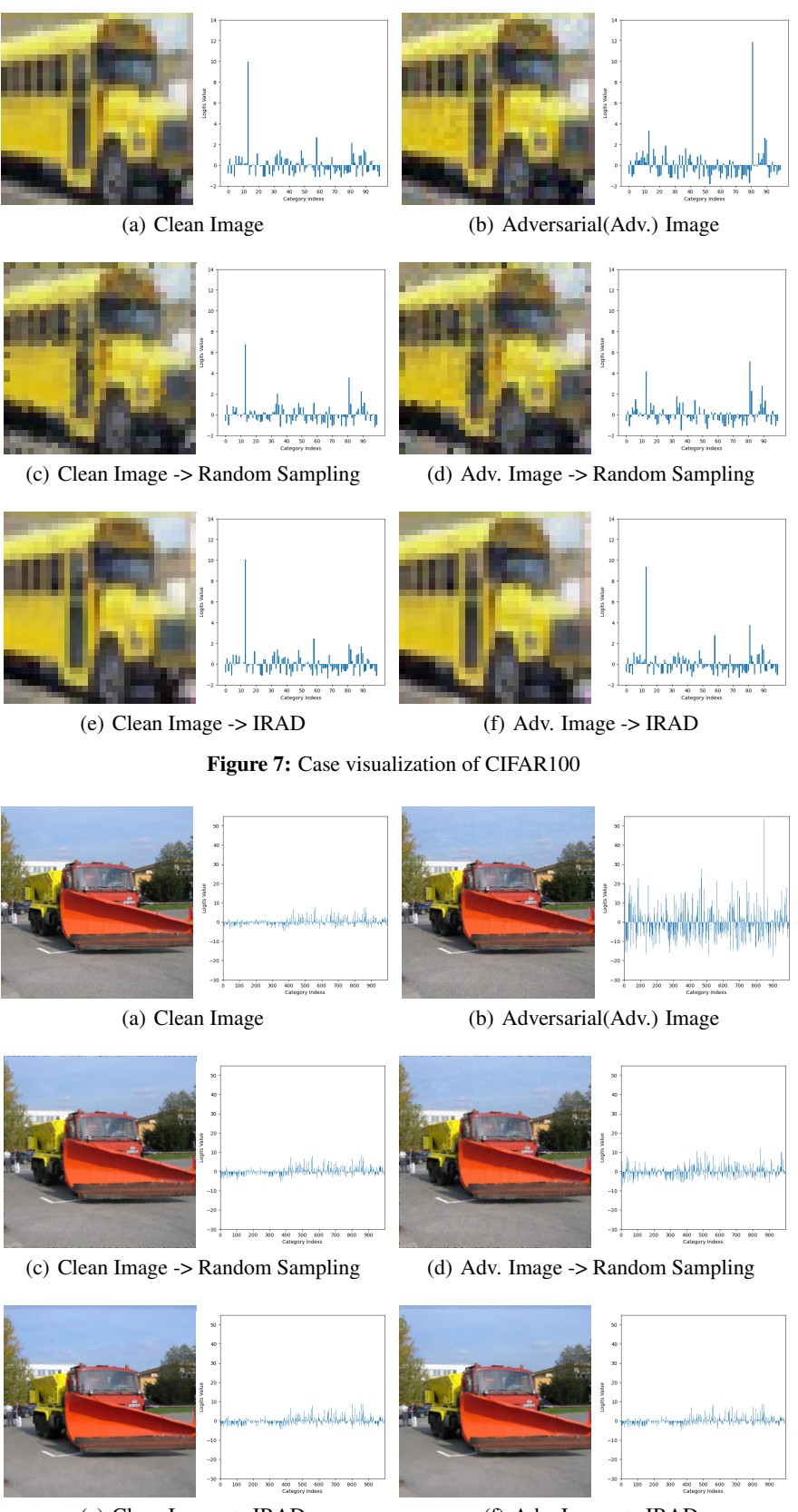

**Figure 7:** Case visualization of CIFAR100

**Figure 8:** Case visualization of ImageNet

