# OpenReview forum: "IRAD: Implicit Representation-driven Image Resampling against Adversarial Attacks"
_ICLR.cc/2024/Conference — ICLR 2024 poster_

### Official Review · Reviewer_mZuj · 2023-10-31

**Soundness:** 3 good
**Presentation:** 4 excellent
**Contribution:** 3 good
**Rating:** 5
**Confidence:** 4

**Summary:**

In this work, the authors adopt image resampling to defend against adversarial attacks. In particular, they first construct an implicit continuous representation for reconstruction. Second, they introduce SampleNet, which automatically generates pixel-wise shifts for resampling. And their method can be extended to the state-of-the-art diffusion-based method, which is significantly accelerated.Extensive experiments on several datasets have shown the effectiveness.

**Strengths:**

1. The paper is well-written and easy to follow.

2. Adopting image resampling is a novel and interesting idea to defend against adversarial attacks.

3. The proposed SampleNet is simple yet effective, which can effectively defend against adversarial attacks to some extent.

4. They have adopted various datasets to validate the effectiveness.

**Weaknesses:**

1. It is not clear why resampling can effectively defend adversarial examples. Especially, image resampling via bilinear or nearest interpolation might be not effective enough to eliminate adversarial perturbation.

2. How can you guarantee that SampleNet is not attacked? As a result, in Table 5, IRAD cannot exhibit robustness when SampleNet is attacked simultaneously.

3. In my opinion, such a resampling method cannot effectively defend against white-box attacks. However, it might be effective in defending against black-box attacks, especially transfer-based attacks [1,2,3]. I suggest you add such a comparison.

4. Since image resampling pre-processes the input image before the model, it is similar to a purifier that eliminates adversarial perturbation [4]. I think it is necessary to compare with such baselines.

[1]  Zhang et al. Patch-wise attack for fooling deep neural network. ECCV 2020.

[2] Wang et al. Enhancing the transferability of adversarial attacks through variance tuning. CVPR 2021.

[3] Wang et al. Admix: Enhancing the transferability of adversarial attacks. ICCV 2021.

[4] Naseer et al. A self-supervised approach for adversarial robustness. CVPR 2020.

**Questions:**

See weakness

---

> ### Author Response · Authors · 2023-11-21
> **Response to Reviewer mZuj (1/4)**
>
> **W1. It is not clear why resampling can effectively defend adversarial examples. Especially, image resampling via bilinear or nearest interpolation might be not effective enough to eliminate adversarial perturbation.**
>
> Thank you for the comments. We propose to leverage image resampling to enhance the adversarial robustness of DNNs against adversarial perturbations. To achieve this goal, we need to address two key questions: 1. how to construct the clean continuous scene based on the input adversarial example? 2. how to estimate the pixel shifting that can further eliminate the influence of adversarial perturbation effectively? **Our method can enhance adversarial robustness significanly since we properly address the two key questions:** First, we build the implicit continuous representation for the first question, which enables us to eliminate the adversarial perturbation while preserving or reconstructing clean details. The reconstruction methods based on bilinear and near interpolations do not have such a capability, thus is not effective to eliminate the adversarial perturbations. Second, we introduce SampleNet, which automatically generates pixel-wise shifts for resampling in response to different inputs, making the reconstructed image have similar semantic features with ground truth category.
>
> To better understand our method, we clarify the effectiveness from two perspectives:
>
> ### A. The effectiveness of implicit representation reconstruction
>
> In this part, we employ four metrics to assess the reconstruction of the implicit representation: PSNR [1], SSIM [2], loss of low frequency and high frequency FFT (Fast Fourier Transform) [3], and LPIPS [4].
>
> 1. PSNR: As shown in Table 4.1a, the PSNR value is higher for implicit representation reconstructed images (36.01) compared to adversarial images (32.1). This indicates that the reconstructed adversarial images have higher fidelity and less distortion compared to the clean images after reconstruction.
> 2. SSIM: The SSIM value is higher for reconstructed adversarial images (0.9973) compared to adversarial images (0.9936). This suggests that the reconstructed adversarial images better preserve the structural details of the original images.
> 3. FFT: Both FFT Low Loss and FFT High Loss are lower for reconstructed adversarial images compared to clean images. This implies that less information is lost in frequency domains during the reconstruction of adversarial images.
> 4. LPIPS: The LPIPS value is significantly lower for reconstructed adversarial images (0.0169) compared to adversarial images (0.0771). This suggests that the reconstructed adversarial images are more perceptually similar to the original images compared to the adversarial images.
>
> Overall, the table suggests that the reconstructed versions of adversarial images tend to exhibit better quality and closer resemblance to the original images across multiple metrics. This also implies that the implicit representation does not sacrifice meaningful information in the raw image when eliminating adversarial perturbations. This demonstrates the effectiveness of using implicit continuous representation to represent images within a continuous coordinate space.
> In Bilinear and Nearest RECONS methods, they rely on the original pixel point as their starting point for representation, meaning they can still be affected by the adversarial patterns present in the adversarial images.
>
> Table 4.1a: Image quality results on CIFAR 10.
>
> |                        | Clean Images and Adversarial Images | Clean Images and Reconstructed Adversarial Images |
> | ---------------------- | :-----------------------------------:| :-------------------------------------------------: |
> | PSNR                   | 32.10                                | 36.01                                             |
> | SSIM                   | 0.9936                              | 0.9973                                            |
> | Low Frequency FFT Loss | 0.0626                              | 0.0506                                            |
> | High Frequency FFT Loss | 0.3636                              | 0.2489                                            |
> | LPIPS                  | 0.0771                              | 0.0169                                            |
>
> [1] Hore, Alain, and Djemel Ziou. "Image quality metrics: PSNR vs. SSIM." 2010 20th international conference on pattern recognition. IEEE, 2010.
>
> [2] Wang, Zhou, et al. "Image quality assessment: from error visibility to structural similarity." IEEE transactions on image processing 13.4 (2004): 600-612.
>
> [3] Cooley, James W., and John W. Tukey. "An algorithm for the machine calculation of complex Fourier series." Mathematics of computation 19.90 (1965): 297-301.
>
> [4] Zhang, Richard, et al. "The unreasonable effectiveness of deep features as a perceptual metric." Proceedings of the IEEE conference on computer vision and pattern recognition. 2018.

---

> ### Author Response · Authors · 2023-11-21
> **Response to Reviewer mZuj (2/4)**
>
> **W1. It is not clear why resampling can effectively defend adversarial examples. Especially, image resampling via bilinear or nearest interpolation might be not effective enough to eliminate adversarial perturbation.**
>
> ### B. The effectiveness of SampleNet
>
> We introduce two additional perspectives to analyze our method and observe the following: First, SampleNet significantly diminishes semantic disparity, bringing reconstructed adversarial and clean representations into closer alignment. Second, SampleNet aligns the decision paths of reconstructed adversarial examples with those of clean examples.
>
> Firstly, we delve into analyzing the features to comprehend how the semantics of the images evolve and perform PCA to visualize the analysis. We randomly selected 100 images from one class and obtained the features of clean images (Clean), adversarial images (Adv), adversarial images after implicit reconstruction (RECONS(Adv)), and adversarial images after IRAD (IRAD(Adv)). We computed the mean in each dimension of the features of clean images to establish the central point of these clean features. Subsequently, we computed the Euclidean distance between the features of other images and the center of clean images. As shown in Table 4.1b, the average Euclidean distance of adversarial image features is significantly distant from clean image features. After the reconstruction of implicit representation, the adversarial image features become closer to the center of clean images' features. Following IRAD, the distance becomes even closer.
> The PCA findings also illustrate a comparable phenomenon. As depicted in Figure 5 (Appendix A.8), clean adversarial images are notably distant from clean images. Following the reconstruction of implicit representation, there are fewer samples situated far from clean images. Moreover, significantly fewer images lie outside the clean distribution after employing IRAD.
> Therefore, this demonstrates that the shift map generated by SampleNet can notably reduce semantic disparity and bring adversarial and clean representations into closer alignment.
>
> Table 4.1b: The comparison of feature distances.
>
> | Comparison            | Euclidean Distance |
> | --------------------- | :------------------: |
> | Clean and Adv         | 2.91               |
> | Clean and RECONS(Adv) | 1.21               |
> | Clean and IRAD(Adv)   | 0.75               |
>
>
>
> Secondly, we follow the methodology of previous research to conduct decision rationale analysis through decision path analysis [5][6][7][8], aiming to understand the decision-making process following resampling. These studies highlight "decision paths" in neural networks, formed by critical nodes like highly contributive neurons at each layer. These paths reflect the network's semantic information flow for specific data sets. When decision paths of two data groups are similar, it indicates closely aligned decision rationales and semantics, revealing how the network interprets and identifies different data. Thus, we obtain the decision paths of Adv, RECONS(Adv), and IRAD(Adv), and calculate their cosine similarity with the decision path of clean images. As shown in Table 4.1c, after the reconstruction of implicit representation, the decision path becomes more similar. By using IRAD, the similarity (i.e. 0.9987) becomes even higher. These results demonstrate that after resampling, our method exhibits a closer alignment with the decision rationales and semantics observed in clean images.
>
> Table 4.1c: Decision path comparison.
>
> | Comparison            | Similarity |
> | --------------------- | :----------: |
> | Clean and Adv         | 0.5482     |
> | Clean and RECONS(Adv) | 0.9915     |
> | Clean and IRAD(Adv)   | 0.9987     |
>
> These two experiments further highlight the effectiveness of our resampling technique in disrupting adversarial textures and mitigating adversarial attacks.
>
> We have added the experimental results and analysis into our Appendix A.7 and Appendix A.8.
>
> [5] Khakzar, Ashkan, et al. "Neural response interpretation through the lens of critical pathways." Proceedings of the IEEE/CVF conference on computer vision and pattern recognition. 2021.
>
> [6] Li, Tianlin, et al. "Understanding adversarial robustness via critical attacking route." Information Sciences 547 (2021): 568-578.
>
> [7] Wang, Yulong, et al. "Interpret neural networks by identifying critical data routing paths." proceedings of the IEEE conference on computer vision and pattern recognition. 2018.
>
> [8] Qiu, Yuxian, et al. "Adversarial defense through network profiling based path extraction." Proceedings of the IEEE/CVF Conference on Computer Vision and Pattern Recognition. 2019.

---

> ### Author Response · Authors · 2023-11-21
> **Response to Reviewer mZuj (3/4)**
>
> **W2. How can you guarantee that SampleNet is not attacked? As a result, in Table 5, IRAD cannot exhibit robustness when SampleNet is attacked simultaneously.**
>
> Thank you for the comments. SampleNet could also be attacked when attackers have full knowledge of defense methods and models. We have discussed this in the original submission in Sec.6.1 with Table 5. Specifically, our initially designed IRAD exhibits vulnerability to white-box attacks while Diffpure exhibits strong robustness under such settings. However, DiffPure’s high performance comes at the expense of significant computational resources. Therefore, we suggest a hybrid approach that capitalizes on the strengths of both DiffPure and IRAD. When we combined IRAD with DiffPure (See Sec. 5), our IRAD+DiffPure (t=20) method achieved comparable SA and RA with DiffPure, and it’s about five times faster.
> We also conduct experiments to explore the integration of IRAD with various steps of DiffPure in Table 4.2. The results show that the hybrid surpasses DiffPure (t=100) in terms of SA, RA, Avg., and time efficiency, demonstrating significant improvements.
>
> Table 4.2: Comparing different steps in the integration of IRAD and DiffPure.
>
> |                      | SA    | RA    | Avg.  | Cost (ms) |
> | -------------------- | :-----: | :-----: | :-----: | :---------: |
> | DiffPure (t=100)            | 89.73 | 75.12 | 82.42 | 132.80     |
> | DISCO                | 89.25 | 0     | 44.63 | 0.38      |
> | IRAD                 | 91.70 | 0     | 45.85 | 0.68      |
> | DiffPure (t=20)      | 93.66 | 8.01  | 50.83 | 27.25     |
> | IRAD+DiffPure (t=20) | 93.42 | 74.05 | 83.73 | 27.70     |
> | DiffPure (t=25)      | 93.55 | 18.28 | 55.92 | 31.93     |
> | IRAD+DiffPure (t=25) | 91.42 | 77.71 | 84.57 | 32.41     |
> | DiffPure (t=30)      | 92.85 | 36.28 | 64.57 | 37.62     |
> | IRAD+DiffPure (t=30) | 92.33 | 82.85 | 87.59 | 38.37     |
> | DiffPure (t=35)      | 93.50  | 48.75 | 71.13 | 44.18     |
> | IRAD+DiffPure (t=35) | 89.14 | 84.57 | 86.86 | 44.86     |
> | DiffPure (t=40)      | 93.71 | 59.42 | 76.57 | 51.78     |
> | IRAD+DiffPure (t=40) | 90.57 | 86.00    | 88.29 | 52.33     |
>
> **W3. In my opinion, such a resampling method cannot effectively defend against white-box attacks. However, it might be effective in defending against black-box attacks, especially transfer-based attacks [1,2,3]. I suggest you add such a comparison.**
>
> **[1] Zhang et al. Patch-wise attack for fooling deep neural network. ECCV 2020.**
>
> **[2] Wang et al. Enhancing the transferability of adversarial attacks through variance tuning. CVPR 2021.**
>
> **[3] Wang et al. Admix: Enhancing the transferability of adversarial attacks. ICCV 2021.**
>
> Thank you for your valuable comments and for providing the meaningful attack methods. We have followed your advice and conducted experiments on the two most recent works [1][2], which include three attack methods (i.e., VmiFgsm, VniFgsm, and Admix). Our results demonstrate that our method significantly outperforms DISCO when facing all three attacks. In comparison to DiffPure, IRAD exhibits notably superior SA and RA when subjected to the VmiFgsm and VniFgsm attacks. While DiffPure slightly surpasses IRAD in RA under the Admix attack, it is worth mentioning that the inference process of DiffPure is approximately 200 times slower than IRAD as shown in Table 5 in our manuscript.
>
> We have added the experiments into our manuscripts (Appendix A.4) and the other attack the reviewer mentioned into the related work.
>
> Table 4.3a: WRN28-10 against VmiFgsm attack on CIFAR10.
>
> | Defense          | SA    | RA    | Avg.  |
> | ---------------- | :-----: | :-----: | :-----: |
> | No Defense       | 94.77 | 0     | 47.39 |
> | DISCO            | 89.24 | 50.51 | 69.88 |
> | DiffPure (t=100) | 89.21 | 85.55 | 87.38 |
> | IRAD             | 91.70  | 86.22 | 88.96 |
>
>
>
> Table 4.3b: WRN28-10 against VniFgsm attack on CIFAR10.
>
> | Defense          | SA    | RA    | Avg.  |
> | ---------------- | :-----: | :-----: | :-----: |
> | No Defense       | 94.77 | 0     | 47.39 |
> | DISCO            | 89.24 | 52.76 | 71.00    |
> | DiffPure (t=100) | 89.21 | 85.49 | 87.35 |
> | IRAD             | 91.70  | 87.65 | 89.68 |
>
> Table 4.3c: WRN28-10 against Admix attack on CIFAR10.
>
> | Defense          | SA    | RA    | Avg.  |
> | ---------------- | :-----: | :-----: | :-----: |
> | No Defense       | 94.77 | 3.82  | 49.30  |
> | DISCO            | 89.24 | 65.91 | 77.58 |
> | DiffPure (t=100) | 89.21 | 85.23 | 87.22 |
> | IRAD             | 91.70  | 82.39 | 87.05 |
>
>
> [1] Wang, Xiaosen, and Kun He. "Enhancing the transferability of adversarial attacks through variance tuning." Proceedings of the IEEE/CVF Conference on Computer Vision and Pattern Recognition. 2021.
>
> [2] Wang, Xiaosen, et al. "Admix: Enhancing the transferability of adversarial attacks." Proceedings of the IEEE/CVF International Conference on Computer Vision. 2021.

---

> ### Author Response · Authors · 2023-11-21
> **Response to Reviewer mZuj (4/4)**
>
> **W4 Since image resampling pre-processes the input image before the model, it is similar to a purifier that eliminates adversarial perturbation [4]. I think it is necessary to compare with such baselines.**
>
> **[4] Naseer et al. A self-supervised approach for adversarial robustness. CVPR 2020.**
>
> Thank you for providing the baseline methods. We conduct evaluation on NRP [1] under the AutoAttack and BPDA attack. Compared with other defense methods, NRP is a purifier network that tries to minimize the perceptual feature difference between clean and Self Supervised Perturbations (SSP) generated adversary. As shown in their work, they use NRP trained on MS-COCO dataset to successfully defend ImageNet models.
>
> Table 4.4: Results of NRP under AutoAttack and BPDA.
>
> | Dataset  | Attack     | SA    | RA   | Avg.  |
> | -------- | ---------- | :-----: | :----: | :-----: |
> | cifar10  | AutoAttack | 92.89 | 3.82 | 46.58 |
> |          | BPDA       | 92.89 | 0.27 | 48.36 |
> | cifar100 | AutoAttack | 77.05 | 6.82 | 41.94 |
> |          | BPDA       | 77.05 | 0.59 | 38.82 |
> | Imagenet | AutoAttack | 72.52 | 20.40 | 46.46 |
> |          | BPDA       | 72.52 | 0.02 | 36.27 |
>
> Our analysis reveals that, in comparison with our method and other baseline approaches, the NRP demonstrates lower effectiveness, particularly in terms of its Robust Accuracy (RA) performance. We think this might be caused by NRP's emphasis on cross-task protection, which encompasses areas such as segmentation, object detection, and similar tasks.
>
> We have added the experiments into our manuscript (Section 6.1).
>
> [1] Naseer, Muzammal, et al. "A self-supervised approach for adversarial robustness." Proceedings of the IEEE/CVF Conference on Computer Vision and Pattern Recognition. 2020.

---

> > ### Comment · Reviewer_mZuj · 2023-11-22
> > **Thanks for the response**
> >
> > I appreciate the authors' response. I think the authors have addressed several concerns. However, considering the robustness against transfer-based attacks, such as Admix, I think it needs a more thorough evaluation to validate the effectiveness of their method. Thus, I maintain my initial score but it is ok for me if this paper is accepted.

---

> > > ### Author Response · Authors · 2023-11-22
> > > **Response to Reviewer mZuj**
> > >
> > > Thank you for your reply. We are glad to know that we have addressed your concerns. Under the Admix attack, there's only a 0.19% difference in average performance (87.05 vs. 87.22). Importantly, IRAD is approximately 200 times faster than Diffpure, highlighting its significant efficiency advantage. Moreover, our method consistently outperforms all baselines in defending against various other attacks, demonstrating its robustness and effectiveness.
> > >
> > > We conducted more experiments to evaluate DiffPure's performance within a shorter timeframe. As the tables show, DiffPure (t=1), which is about 3 times slower than IRAD, has almost no ability to defend against the attacks. IRAD significantly outperforms DiffPure (t=20) and is about 40 times faster.
> > >
> > > We kindly request that you consider the efficiency aspect when determining the final grade. Time efficiency holds significant importance in real-world implementation, particularly in scenarios where high time efficiency is demanded.
> > >
> > > We have included the defense time cost comparisons in the tables to facilitate comparisons.
> > >
> > > Table 4.5a: WRN28-10 against VmiFgsm attack on CIFAR10.
> > >
> > > | Defense          | SA    | RA    | Avg.  | Cost (ms) |
> > > | ---------------- | ----- | ----- | ----- | --------- |
> > > | No Defense       | 94.77 | 0     | 47.39 | 0         |
> > > | DISCO            | 89.24 | 50.51 | 69.88 | 0.38      |
> > > | DiffPure (t=1)   | 94.63 | 0     | 47.32 | 2.43      |
> > > | DiffPure (t=20)  | 94.12 | 14.36 | 54.24 | 27.25     |
> > > | DiffPure (t=100) | 89.21 | 85.55 | 87.38 | 132.80     |
> > > | IRAD             | 91.70  | 86.22 | 88.96 | 0.68      |
> > >
> > >
> > >
> > > Table 4.5b: WRN28-10 against VniFgsm attack on CIFAR10.
> > >
> > > | Defense          | SA    | RA    | Avg.  | Cost (ms) |
> > > | ---------------- | ----- | ----- | ----- | --------- |
> > > | No Defense       | 94.77 | 0     | 47.39 | 0         |
> > > | DISCO            | 89.24 | 52.76 | 71.00    | 0.38      |
> > > | DiffPure (t=1)   | 94.63 | 0.01  | 47.32 | 2.43      |
> > > | DiffPure (t=20)  | 94.12 | 13.52 | 53.82 | 27.25     |
> > > | DiffPure (t=100) | 89.21 | 85.49 | 87.35 | 132.80     |
> > > | IRAD             | 91.70  | 87.65 | 89.68 | 0.68      |
> > >
> > >
> > >
> > > Table 4.5c: WRN28-10 against Admix attack on CIFAR10.
> > >
> > > | Defense          | SA    | RA    | Avg.  | Cost (ms) |
> > > | ---------------- | ----- | ----- | ----- | --------- |
> > > | No Defense       | 94.77 | 3.82  | 49.3  | 0         |
> > > | DISCO            | 89.24 | 65.91 | 77.58 | 0.38      |
> > > | DiffPure (t=1)   | 94.63 | 4.15  | 49.39 | 2.43      |
> > > | DiffPure (t=20)  | 94.11 | 33.78 | 63.95 | 27.25     |
> > > | DiffPure (t=100) | 89.21 | 85.23 | 87.22 | 132.80     |
> > > | IRAD             | 91.70  | 82.39 | 87.05 | 0.68      |

---

> > > ### Author Response · Authors · 2023-11-23
> > > **Response to Reviewer mZuj**
> > >
> > > To further address your concerns about IRAD's performance against transfer-based attacks, we conducted more experiments using ImageNet. Please note that we followed the approach outlined in the DiffPure paper, setting t=150 when dealing with ImageNet.
> > >
> > > The tables indicate that IRAD outperforms DISCO significantly across all three attack scenarios. Furthermore, not only does IRAD surpass Diffpure in performance under all three attacks, but it is also more than 100 times faster than Diffpure. This demonstrates IRAD's superior performance and time efficiency.
> > >
> > >
> > > Table 4.6a: ResNet50 against VmiFgsm attack on ImageNet.
> > >
> > > | Defense    | SA    | RA    | Avg.  | Cost (ms) |
> > > | ---------- | ----- | ----- | ----- | --------- |
> > > | No Defense | 76.72 | 0     | 38.36 | 0         |
> > > | DISCO      | 72.66 | 52.98 | 62.82 | 25.16     |
> > > | DiffPure   | 69.42 | 64.90  | 67.16 | 4098.68   |
> > > | IRAD       | 72.14 | 71.35 | 71.75 | 37.78     |
> > >
> > >
> > >
> > > Table 4.6b: ResNet50 against VniFgsm attack on ImageNet.
> > >
> > > | Defense          | SA    | RA    | Avg.  | Cost (ms) |
> > > | ---------------- | ----- | ----- | ----- | --------- |
> > > | No Defense       | 76.72 | 0.04  | 38.38 | 0         |
> > > | DISCO            | 72.66 | 52.76 | 62.71 | 25.16     |
> > > | DiffPure | 69.42 | 64.22 | 66.82 | 4098.68   |
> > > | IRAD             | 72.14 | 71.4  | 71.77 | 37.78     |
> > >
> > >
> > >
> > > Table 4.6c: ResNet50 against Admix attack on ImageNet.
> > >
> > > | Defense          | SA    | RA    | Avg.  | Cost (ms) |
> > > | ---------------- | ----- | ----- | ----- | --------- |
> > > | No Defense       | 76.72 | 3.32  | 40.02 | 0         |
> > > | DISCO            | 72.66 | 42.74 | 57.7  | 25.16     |
> > > | DiffPure | 69.42 | 61.34 | 65.38 | 4098.68   |
> > > | IRAD             | 72.14 | 63.8  | 67.97 | 37.78     |
> > >
> > >
> > > Thank you once more for dedicating your time and effort to reviewing our paper! As the discussion phase nears its end, we're eager to confirm if we've successfully addressed the concern you raised. If you find that this concern has been addressed, we would greatly appreciate it if you could reflect this in your paper rating.

---

### Official Review · Reviewer_4nS8 · 2023-11-02

**Soundness:** 3 good
**Presentation:** 2 fair
**Contribution:** 3 good
**Rating:** 6
**Confidence:** 3

**Summary:**

This paper introduces a test-time adversarial defense mechanism realized via image resampling. Central to their proposal is the "Implicit Representation Driven Image Resampling (IRAD)" methodology, which operates in two distinct phases: a) the creation of a continuous coordinate space, and b) the utilization of the newly proposed SampleNet. Unlike conventional heuristic sampling strategies, SampleNet is designed to predict pixel-wise shifts based on the embedding of the input image automatically. Moreover, in a pursuit of optimal performance and a marked increase in processing speed, the authors have integrated their method with DiffPure.

**Strengths:**

1. The introduced image-resampling defense "IRAD" substantially enhances the Robust Accuracy (RA) when juxtaposed with the elementary implementation of image resampling (IR).
2. The paper presents SampleNet, a novel approach for automated sampling, designed to supplant traditional heuristic sampling strategies.
3. The methodology presented has been rigorously evaluated across various datasets and neural network architectures.

**Weaknesses:**

1. There's room for optimization in the placement of tables and figures to enhance readability and presentation.
2. Merging IRAD with a 20-step DiffPure yields superior results while economizing computation time. Nonetheless, a more comprehensive analysis examining the integration of IRAD with varying steps of DiffPure would enrich the study's depth and relevance.

**Questions:**

None

---

> ### Author Response · Authors · 2023-11-21
> **Response to Reviewer 4nS8**
>
> **W1. There's room for optimization in the placement of tables and figures to enhance readability and presentation.**
>
> Thank you for your guidance. We have meticulously refined the arrangement of figures and tables, making concerted efforts to ensure their proximity to the corresponding text in our manuscript.
>
> **W2. Merging IRAD with a 20-step DiffPure yields superior results while economizing computation time. Nonetheless, a more comprehensive analysis examining the integration of IRAD with varying steps of DiffPure would enrich the study's depth and relevance.**
>
> Thank you for suggesting a meaningful comparison experiment.
>
> We conduct additional experiments on integrating IRAD with various steps of DiffPure. The updated results, along with the previous findings, are presented in Table 3.1. Due to the significant time and resource costs associated with implementing DiffPure, we perform the experiment on 300 randomly selected images from the test set. The findings indicate that IRAD consistently enhances DiffPure's performance across various steps. For example, with t=40, the combination of IRAD and DiffPure (t=40) notably increases the Robust Accuracy (RA) of DiffPure (t=40) from 59.42 to 86.00. Moreover, it surpasses DiffPure (t=100) in terms of SA, RA, Avg., and time efficiency, demonstrating significant improvements.
>
> We have added the experimental results to Appendix A.5.
>
> Table 3.1 Comparing different steps in the integration of IRAD and DiffPure.
>
> |                      | SA    | RA    | Avg.  | Cost (ms) |
> | -------------------- | :-----: | :-----: | :-----: | :---------: |
> | DiffPure             | 89.73 | 75.12 | 82.42 | 132.80     |
> | DISCO                | 89.25 | 0     | 44.63 | 0.38      |
> | IRAD                 | 91.70 | 0     | 45.85 | 0.68      |
> | DiffPure (t=20)      | 93.66 | 8.01  | 50.83 | 27.25     |
> | IRAD+DiffPure (t=20) | 93.42 | 74.05 | 83.73 | 27.70     |
> | DiffPure (t=25)      | 93.55 | 18.28 | 55.92 | 31.93     |
> | IRAD+DiffPure (t=25) | 91.42 | 77.71 | 84.57 | 32.41     |
> | DiffPure (t=30)      | 92.85 | 36.28 | 64.57 | 37.62     |
> | IRAD+DiffPure (t=30) | 92.33 | 82.85 | 87.59 | 38.37     |
> | DiffPure (t=35)      | 93.50  | 48.75 | 71.13 | 44.18     |
> | IRAD+DiffPure (t=35) | 89.14 | 84.57 | 86.86 | 44.86     |
> | DiffPure (t=40)      | 93.71 | 59.42 | 76.57 | 51.78     |
> | IRAD+DiffPure (t=40) | 90.57 | 86.00    | 88.29 | 52.33     |

---

### Official Review · Reviewer_ynQw · 2023-11-02

**Soundness:** 3 good
**Presentation:** 3 good
**Contribution:** 2 fair
**Rating:** 6
**Confidence:** 4

**Summary:**

This paper proposed a new method for defending against adversarial attacks. Given an input image, the proposed method aims to first build an implicit representation and then reconstruct the image by resampling. For further improvement, the paper also proposes to introduce a network to dispatch different inputs to different level pixel-wise shifts for resampling. Experiments results show that the proposed methods improve the adversarial robustness.

**Strengths:**

- The underlying idea and the proposed method appear very novel to me.

- The paper conducts extensive experiments and demonstrates significant improvements over the baseline.

- The paper is well-written and easy to read

**Weaknesses:**

I am having difficulty being convinced of why this approach works. The paper's objective is to construct an implicit representation for the input image and then reconstruct the clean image from this implicit representation, claiming that the reconstructed image would be free from any adversarial patterns. However, if we assume that the underlying implicit representation faithfully represents the input signal, it should also contain the adversarial patterns that present in the input signal. This contradicts the explanation for why the proposed method is effective.

To the best of my thoughts, the only plausible explanation to explain why the proposed method works well quantitatively is that the underlying implicit representation is biased toward fitting low-frequency or smooth signals, and thus, it fails to represent high-frequency adversarial patterns. However, this would imply a degradation in the image quality represented in the implicit model, as it suggests that the model cannot faithfully represent high-frequency signals. In either case, this highlights certain weaknesses in the proposed method

**Questions:**

See my comments on the weaknesses session above. I'm willing to improve my rating if the author can convince me in the rebuttal period.

---

> ### Author Response · Authors · 2023-11-21
> **Response to Reviewer ynQw (1/2)**
>
> ## W1. Why the proposed method works?
>
> **I am having difficulty being convinced of why this approach works. The paper's objective is to construct an implicit representation for the input image and then reconstruct the clean image from this implicit representation, claiming that the reconstructed image would be free from any adversarial patterns. However, if we assume that the underlying implicit representation faithfully represents the input signal, it should also contain the adversarial patterns that present in the input signal. This contradicts the explanation for why the proposed method is effective.**
>
> Thank you for the valuable comments. The reviewer might have misunderstood the idea and certain details of this work. To clarify, we will address the question from two aspects: the main idea and why the proposed method works.
>
> **The main idea:** We propose to leverage image resampling to enhance the adversarial robustness of DNNs against adversarial perturbations. To achieve this goal, we need to address two key questions: 1. how to construct the clean continuous scene based on the input adversarial example? 2. how to estimate the pixel shifting that can further eliminate the influence of adversarial perturbation effectively? *Our method achieves the objective of enhancing adversarial robustness since we properly address the two key questions.*
>
> **Why proposed method works:**
>
> 1.
> We address the first question properly by training a local implicit representation (i.e., Eq. (9) in Sec. 4.2) that can estimate the clean color of arbitrary given coordinates of a pixel. *Note that the implicit representation is not designed to faithfully represent the input signal but to eliminate adversarial perturbations and reconstruct the clean details.*
> To achieve this, when given the coordinates of a pixel, we utilize an MLP to deduce the pixel's true color by considering the deep features of its neighboring pixels and their spatial distances. We train the MLP and the feature extraction model through the adversarial denoising task (refer to the discussion in Sec. 6.3 with Table 7).
> During the inference phase, the well-trained MLP can eliminate the adversarial noise while preserving the clean details. As illustrated in the visualization results (i.e., Figure 6-8) in the appendix, most of the adversarial perturbations are removed (for further explanations, please refer to the explanation section).
>
> To further validate this, we add an image quality experiment. Specifically, we calculate the PSNR [1] and SSIM [2] between the clean image and adversarial image, the clean image and reconstructed adversarial image on CIFAR10 dataset, respectively. As shown in Table 2.1, the adversarial perturbations are significantly eliminated:
> (a) As shown in Table 2.1, after implicit representation reconstruction, the PSNR value increases from 32.1 to 36.01. This indicates that the reconstructed adversarial images have higher fidelity and less distortion compared to the clean images after reconstruction.
> (b) We have similar observations on the SSIM scores. This suggests that the reconstructed adversarial images better preserve the structural details of the original images.
>
>
> Table 2.1: Image quality results on CIFAR 10.
>
> |                        | Clean Images and Adversarial Images | Clean Images and Reconstructed Adversarial Images |
> | ---------------------- | :-----------------------------------: | :-------------------------------------------------: |
> | PSNR                   | 32.10                                | 36.01                                             |
> | SSIM                   | 0.9936                              | 0.9973
> | LPIPS                  | 0.0771                              | 0.0169                                                         |
> | Low Frequency FFT Loss | 0.0626                              | 0.0506                                            |
> | High Frequency FFT Loss | 0.3636                              | 0.2489                                            |
>
>
> 2. We address the second question properly by training a SampleNet that can predict the shifting that further breaks the influence of adversarial patterns and pushes the semantic features of the input image towards the category center. Please refer to the response to the first reviewer (i.e., x2bE) for details.

---

> ### Author Response · Authors · 2023-11-21
> **Response to Reviewer ynQw (2/2)**
>
> ## W2. How to explain the effectiveness?
>
> **To the best of my thoughts, the only plausible explanation to explain why the proposed method works well quantitatively is that the underlying implicit representation is biased toward fitting low-frequency or smooth signals, and thus, it fails to represent high-frequency adversarial patterns. However, this would imply a degradation in the image quality represented in the implicit model, as it suggests that the model cannot faithfully represent high-frequency signals. In either case, this highlights certain weaknesses in the proposed method**
>
> Thank you for contributing a new perspective to understand our method. As previously discussed, one main reason for the effectiveness is that the implicit representation could effectively eliminate the adversarial perturbation while reconstructing clean details very well. While comprehending this process through the adjustment of low-frequency components and the elimination of high-frequency signals may not suffice to elucidate all the outcomes, we endeavor to provide an explanation, as it remains a constructive approach to enhance the clarity of the paper. In the frequency domain, our method is not to remove all high-frequency signals uniformly but to eliminate the high-frequency signals caused by adversarial perturbations and preserve the ones of clean/raw information.
>
> For additional clarification, we extend the experiment in W1 to the frequency domain and employ two metrics to assess the reconstruction of the implicit representation: loss of low frequency and high frequency FFT (Fast Fourier Transform) [3], and LPIPS [4].
>
> 1. FFT: Both FFT Low Loss and FFT High Loss are lower for reconstructed adversarial images compared to clean images. This implies that less information is lost in frequency domains during the reconstruction of adversarial images.
>
> 2. LPIPS: The LPIPS value is significantly lower for reconstructed adversarial images (0.0169) compared to adversarial images (0.0771). This suggests that the reconstructed adversarial images are more perceptually similar to the original images compared to the adversarial images.
>
>
> Overall, Table 2.1 suggests that the reconstructed versions of adversarial images tend to exhibit better quality and closer resemblance to the original images across multiple metrics. This implies that the implicit representation does not sacrifice useful high-frequency information when eliminating adversarial perturbations. This demonstrates the effectiveness of using implicit continuous representation to represent images within a continuous coordinate space.
>
> We have added the experimental results and analysis to Appendix A.7.
>
> [1] Hore, Alain, and Djemel Ziou. "Image quality metrics: PSNR vs. SSIM." 2010 20th international conference on pattern recognition. IEEE, 2010.
>
> [2] Wang, Zhou, et al. "Image quality assessment: from error visibility to structural similarity." IEEE transactions on image processing 13.4 (2004): 600-612.
>
> [3] Cooley, James W., and John W. Tukey. "An algorithm for the machine calculation of complex Fourier series." Mathematics of computation 19.90 (1965): 297-301.
>
> [4] Zhang, Richard, et al. "The unreasonable effectiveness of deep features as a perceptual metric." Proceedings of the IEEE conference on computer vision and pattern recognition. 2018.

---

> > ### Comment · Reviewer_ynQw · 2023-11-23
> >
> > Thank you for your thorough comment and I appreciate it. The comment above has, to some extent, addressed my concerns, and I am willing to adjust my rating to acceptance accordingly

---

> > > ### Author Response · Authors · 2023-11-23
> > > **Response to Reviewer ynQw**
> > >
> > > Thank you for your prompt response. We are glad to know that we have addressed your concerns. We truly appreciate your positive and valuable comments, which have greatly contributed to enhancing the quality of our manuscript. We also appreciate your recognition of our work.

---

### Official Review · Reviewer_x2bE · 2023-11-02

**Soundness:** 4 excellent
**Presentation:** 3 good
**Contribution:** 3 good
**Rating:** 8
**Confidence:** 4

**Summary:**

This paper introduces a novel image resampling approach for adversarial attacks. Specifically, two implicit neural representations are adopted to model the image reconstruction and shift map, respectively. With the estimated continuous representation and the learned shifting, the generated image becomes robust toward adversarial attacks. This method achieves significant performance on several benchmarks.

**Strengths:**

1. The proposed method provides a novel approach to solving the adversarial attack problem. This is the first paper to solve the problem in such an implicit representation manner. The idea is impressive。

2. The writing is easy to follow. It starts with the naive solution, and then gradually introduces the proposed one. Each step is logically based on the last step.

3. The experimental results are strong. It not only significantly outperforms baselines in Table 1, but also achieves higher performance than state-of-the-art approaches with totally different solutions.

**Weaknesses:**

1. One important ablation study is lacking. Table 8 verifies the effectiveness of the SampleNet. However, the usefulness of RECONS is not proven. How about the performance of bilinear interpolation with SampletNet?

2. The learned shift map contains explainable information, however, analysis of which is lacking. The only visualization shown in Fig 2 contains little information and a deeper analysis should be adopted.

3. The running time of the proposed method is not provided.

**Questions:**

Are there any severe distortions in the shift map?

---

> ### Author Response · Authors · 2023-11-21
> **Response to Reviewer x2bE (1/3)**
>
> **W1. One important ablation study is lacking. Table 8 verifies the effectiveness of the SampleNet. However, the usefulness of RECONS is not proven. How about the performance of bilinear interpolation with SampletNet?**
>
> Thank you for the helpful advice.
>
> We conducted an ablation study to integrate various RECONS methods with SampleNet. It's important to note that SampleNet is an MLP that takes the features of pixel \[i,j\] and the coordinate values as input, predicting their shifting directly. Since the Nearest and Bilinear RECONS methods could not generate the pixel features as the implicit representations do, we directly utilize the encoder from the implicit representations for a fair comparison. The experimental results are shown in Table 1.1.
>
> Table 1.1: Comparing different reconstruction methods.
>
> | RECONS                  | SA    | RA    | Avg.  |
> | ----------------------- | ----- | ----- | ----- |
> | Nearest                 | 93.46 | 0.96  | 47.21 |
> | Bilinear                | 92.61 | 79.20  | 85.91 |
> | Implicit Representation | 91.70  | 89.72 | 90.71 |
>
>
> As depicted in the table, despite having a high SA, Nearest RECONS lack the capability to defend against adversarial attacks. This limitation arises from its ability to acquire only the pixel values originally sampled from the images. The Bilinear RECONS exhibit superior RA in comparison to the Nearest RECONS and outperform basic image resampling methods under Bilinear RECONS showcased in Table 1 of the paper, highlighting the effectiveness of SampleNet. Compared to these two RECONS, the implicit representation RECONS achieves an even higher RA and Avg., demonstrating their superiority over these two RECONS. Overall, the data from the table suggests that Nearest and Bilinear RECONS are less effective than implicit representation-based methods when integrated with SampleNet.
>
> We have added the discussion to our manuscript (Appendix A.6).

---

> ### Author Response · Authors · 2023-11-21
> **Response to Reviewer x2bE (2/3)**
>
> **W2. The learned shift map contains explainable information, however, analysis of which is lacking. The only visualization shown in Fig 2 contains little information and a deeper analysis should be adopted.**
>
> Thanks for the helpful suggestion.
>
> To explain the effectiveness of the shift map, we here analyze and understand the changes in image semantics when the learned shift map is applied or not.  We approach this by examining the features in the middle layer and exploring the decision rationale behind image classification. This dual-perspective approach helps clarify the shift map's effectiveness.
>
> Firstly, we delve into analyzing the features to comprehend how the semantics of the images evolve and perform PCA to visualize the analysis. We randomly selected 100 images from one category and obtained the features of clean images (Clean), adversarial images (Adv), adversarial images after implicit reconstruction (RECONS(Adv)), and adversarial images after IRAD (IRAD(Adv)). We computed the mean in each dimension of the features of clean images to establish the central point of these clean features. Subsequently, we computed the Euclidean distance between the features of other images and the center of clean images. As shown in Table 1.2a, the average Euclidean distance of adversarial image features is significantly distant from clean image features. After the reconstruction of implicit representation, the adversarial image features become closer to the center of clean images' features. By utilizing IRAD, the distance becomes even closer.
> The PCA findings also illustrate a comparable phenomenon. As depicted in Figure 5 (Appendix A.8), clean adversarial images are notably distant from clean images. After the reconstruction of implicit representation, there are fewer samples situated far from clean images. Moreover, significantly fewer images lie outside the clean distribution after employing IRAD.
> Therefore, this demonstrates that the shift map generated by SampleNet can notably reduce semantic disparity and bring adversarial and clean representations into closer alignment.
>
> Table 1.2a: The comparison of feature distances.
>
> | Comparison            | Euclidean Distance |
> | --------------------- | :------------------: |
> | Clean and Adv         | 2.91               |
> | Clean and RECONS(Adv) | 1.21               |
> | Clean and IRAD(Adv)   | 0.75               |
>
>
>
>
> Secondly, we follow the methodology of previous research to conduct decision rationale analysis through decision path analysis [1][2][3][4], aiming to understand the decision-making process following resampling. These studies highlight "decision paths" in neural networks, formed by critical nodes like highly contributive neurons at each layer. These paths reflect the network's semantic information flow for specific data sets. When the decision paths of two data groups are similar, it indicates closely aligned decision rationales and semantics, revealing how the network interprets and identifies different data. Thus, we obtain the decision paths of Adv, RECONS(Adv), and IRAD(Adv), and calculate their cosine similarity with the decision path of clean images. As shown in Table 1.2b, after the reconstruction of implicit representation, the decision path becomes more similar. By using IRAD, the similarity (i.e. 0.9987) becomes even higher. These results demonstrate that after resampling, our method exhibits a closer alignment with the decision rationales and semantics observed in clean images.
>
> Table 1.2b: The comparison of decision paths.
>
> | Comparison            | Similarity |
> | --------------------- | :----------: |
> | Clean and Adv         | 0.5482     |
> | Clean and RECONS(Adv) | 0.9915     |
> | Clean and IRAD(Adv)   | 0.9987     |
>
>
>
> These two experiments further highlight the effectiveness of our resampling technique in disrupting adversarial textures and mitigating adversarial attacks. The tables of experimental results and the PCA result are included in the Appendix (A.8).
>
> [1] Khakzar, Ashkan, et al. "Neural response interpretation through the lens of critical pathways." Proceedings of the IEEE/CVF conference on computer vision and pattern recognition. 2021.
>
> [2] Li, Tianlin, et al. "Understanding adversarial robustness via critical attacking route." Information Sciences 547 (2021): 568-578.
>
> [3] Wang, Yulong, et al. "Interpret neural networks by identifying critical data routing paths." proceedings of the IEEE conference on computer vision and pattern recognition. 2018.
>
> [4] Qiu, Yuxian, et al. "Adversarial defense through network profiling based path extraction." Proceedings of the IEEE/CVF Conference on Computer Vision and Pattern Recognition. 2019.

---

> ### Author Response · Authors · 2023-11-21
> **Response to Reviewer x2bE (3/3)**
>
> **W3. The running time of the proposed method is not provided.**
>
> Thank you for your comments. In fact, we include the running time cost comparison in Table 5 of our manuscript. The results show that DiffPure takes 132 ms per image, which is time-consuming. Our IRAD+DiffPure (t=20) method achieves comparable SA and RA with DiffPure, and it's about five times faster. It should also be noted that the running time cost statistics are consistent under other adversary scenarios.
>
> **Q1. Are there any severe distortions in the shift map?**
>
> Thank you for your comments. We assess distortions from both visual and semantic perspectives.
>
> As highlighted in the 'More Visualization Results' section of Appendix A.9, our visual examination reveals that the resampled images closely resemble the clean images, with no significant distortions due to the shift map.
>
> Additionally, as detailed in our answer to W2, we investigate the impact of the shift map on the features in the middle layer and the decision-making process in image classification. Our findings demonstrate that our method successfully reconstructs images without causing major semantic distortions. The details of these semantic evaluations are presented in Appendices A.7 and A.8.

---

> > ### Comment · Reviewer_x2bE · 2023-11-23
> > **A possible explanation**
> >
> > All my issues have been solved.
> > But I'm still interested in understanding how rhis works. My understanding is: the learned Interpretation reduces the weights of adversarial pixels, thus leads to the results in Table 1.2 ab. Hope more insight analysis will be done.

---

> > > ### Author Response · Authors · 2023-11-23
> > > **Response to Reviewer x2bE**
> > >
> > > Thank you for your reply. We are delighted to know that we have resolved all your issues. We truly appreciate your constructive and helpful comments, which have significantly contributed to enhancing the quality of our manuscript.
> > >
> > > The angle you mentioned is interesting: explaining our method as a learnable interpretation method. As described in Eq. (9) in the paper, the function φ(·) represents a multilayer perceptron (MLP), and the implicit representation aims to learn how to predict the color of the pixel [u, v] based on the embedding of the pixel [k, l] and their spatial distance. Since the response period is coming to an end, we will be committed to analyzing this point in our future version.
> > >
> > > Thank you once again for your encouraging and insightful feedback; engaging in discussions with you is truly inspiring.

---

### Author Response · Authors · 2023-11-21
**Paper Revision Summary**

# Summary


We appreciate the thoughtful comments and helpful criticism from every reviewer. We are delighted that the reviewers thought our paper was "novel", "interesting", "easy to follow" and "rigorously evaluated".

Below is a summary of our paper update, and we mark the updates in our paper with blue color.

1. We have meticulously refined the arrangement of figures and tables in our manuscript.
2. **[Experimental Results 6.1]** We compare our method with one more baseline method NRP under the AutoAttack and BPDA attack in Table 2 and 3.
3. **[Appendix A.4]** We include three transfer-based attack methods (i.e., VmiFgsm, VniFgsm, and Admix) to evaluate the defense performance.
4. **[Appendix A.5]** We exhibit experiment results on integrating IRAD with various steps of DiffPure.
5. **[Appendix A.6]** We show an ablation study to integrate various RECONS methods with SampleNet.
6. **[Appendix A.7]** We take four metrics to assess and explain the reconstruction by the implicit representation.
7. **[Appendix A.8]**  We introduce two additional perspectives to
analyze our method and observe the effectiveness of SampleNet.
8. **[Related Work]** We discuss more relevant papers in our manuscript.

---

### Meta-Review · Area_Chair_5ArK · 2023-12-13

**Metareview:**

This paper develops a novel image resampling approach for adversarial attacks, introducing two implicit neural representations to model image reconstruction and shift map. Extensive experiments across various datasets and models are provided to support the efficacy of the proposed method.

Overall, reviewers commend the paper's novelty, thorough experimentation, and clarity of writing. The primary concerns raised pertained to the need for additional ablation studies and deeper analysis, which are well addressed in the rebuttal. The final decision is accepted.

In the final version, the authors should include these additional analyses/ablations provided in the rebuttal to enhance the quality of this paper.

**Justification For Why Not Higher Score:**

As mentioned by Reviewer mZuj, this method alone may still be limited in the white-box and transfer-attack settings.

**Justification For Why Not Lower Score:**

The paper provides a novel, efficient, and effective strategy for defending against adversarial attacks, which deserves a poster presentation at NeurIPS for sharing this valuable knowledge with the community.

---

### Decision · Program_Chairs · 2024-01-16

Accept (poster)